# Regularized Maximum Mean Discrepancy for Variable Selection

## Abstract

This paper proposes a variable selection method based on maximum mean discrepancy (MMD) under sparsity. Our approach assigns weights to each variable and optimizes them within a regularized MMD framework, where some weights are pushed to zero, corresponding to variables that are not important. These optimized weights serve as an importance measure for variables contributing to the difference between two distributions. We propose an object-oriented variable selection approach, where the selected variables via the optimized weights also minimize a specified loss function associated with particular objects or tasks. We focus on two common scenarios–two-sample tests and classification–aiming to improve the power of the MMD test and enhance the classification accuracy of classifiers. Theoretical results on the consistency of the estimated weights and the convergence of the accelerated algorithms are established. Simulations and real-data analysis validate the effectiveness of the proposed method.

## 1 Introduction

Two-sample tests, which aim to determine whether two samples are drawn from the same distribution, have been extensively studied. Classical tests for the equality of two distributions, including the Kolmogorov-Smirnov test (Smirnov, 1939) and the Cramér–von Mises test (Anderson, 1962), are only effective for low-dimensional data. Recently, the literature has witnessed the development of many non-parametric testing methods (Rosenbaum, 2005; Székely & Rizzo, 2013; Gretton et al., 2012; Wang et al., 2021) for multi-dimensional or even high-dimensional data. Among these, the maximum mean discrepancy (MMD) test (Gretton et al., 2012), which leverages kernel mean embeddings to quantify the discrepancy between two distributions, has gained significant attention. The MMD test has found applications in various fields, such as generative models (Sutherland et al., 2017; Binkowski et al., 2018), transfer learning (Long et al., 2017; Wei et al., 2019), and change-point detection (Cheng & Xie, 2021). With the rising popularity of the MMD method, researchers have proposed several refinements and enhancements (Sutherland et al., 2017; Liu et al., 2020; Biggs et al., 2023) aimed at improving its test power in high-dimensional and complex data settings, and some researchers have considered variants of the MMD statistic (Zaremba et al., 2013; Chwialkowski et al., 2015; Ramdas et al., 2015a; Shekhar et al., 2022) to accelerate computation.

**Motivation**. In multi-dimensional or high-dimensional data, *sparsity* is a common phenomenon where only a small proportion of *important* variables contribute to the differences between two distributions. Non-important variables, often referred to as noise or irrelevant features, can adversely affect the performance of two-sample tests. As shown in Mueller & Jaakkola (2015), an excessive number of non-important variables may obscure the signals of important variables, making the distributions of the two samples become more similar and, consequently, reducing the power of the MMD test. In addition, the binary classification also faces the issue of a sharp decline in classification accuracy due to a large number of non-important variables. This motivates the great need to identify important variables, also referred to as *variable selection* under sparsity.

**Our approach**. In this paper, we propose a variable selection method for *two-sample problems* under the MMD framework. We first assign a non-negative weight to each variable and obtain the optimal weights by maximizing the MMD statistic. These optimal weights can serve as a reference for variable importance. To prevent overemphasis on variables with strong signals, an $\ell_2$-regularization term is incorporated into the weighted MMD for variable selection. Variables with weights signif-

icantly greater than $0$ are then identified as important. Considering that variable selection is often a preliminary task for other objectives, we propose an object-oriented approach, which selects variables via the optimized weights that minimize a specified loss function tied to specific objects or tasks. Moreover, we introduce an algorithm to accelerate the computation of the optimal weights.

**Related work**. Yamada et al. (2018); Lim et al. (2020) propose performing marginal MMD tests on individual variables to identify important ones, but this approach overlooks interactions and dependencies, failing to capture differences in joint distributions. Similarly, Adamer et al. (2024) ranks variable importance by optimizing feature weights but focus on marginal differences. Wang et al. (2023) select variables by maximizing the MMD estimator to boost test power. However, this method requires specifying a fixed number of variables, $d$, which risks including irrelevant variables or excluding crucial ones if chosen improperly.

The variable selection problem for binary classification has also attracted significant attention. Logistic regression with $\ell_1$-regularization Hastie et al. (2009) is widely used but assumes a specific data model, limiting its applicability when the data deviates from this model. van Reenen et al. (2016); Bénard et al. (2024) propose model-free variable selection methods for binary classification, but these also focus on marginal differences, overlooking dependence in joint distributions.

The selection of an appropriate regularization parameter in the regularization term is crucial for achieving better model fitting. Consequently, methods for choosing the regularization parameter have been extensively studied. A commonly used and effective method for selecting regularization parameters is the cross-validation approach (Stone, 1974; Arlot & Celisse, 2010). In addition, information criterion methods, such as AIC (Akaike, 1974), BIC (Schwarz, 1978) and GIC (Konishi & Kitagawa, 1996) have also been widely applied. In addition, some researchers treat the regularization parameter as a hyperparameter within the model and optimize it dynamically using gradient-based methods (Bengio, 2000; Luketina et al., 2016; Franceschi et al., 2017).

**Contributions**. The main contributions of this paper can be summarized as follows: **(a)** We propose a novel method for identifying important variables responsible for distributional differences in sparse settings. Our approach leverages the advantages of the MMD test, which is non-parametric, model-free, and accounts for dependence among variables. This allows our method to outperform marginal approaches. **(b)** We propose an object-oriented variable selection method tailored to specific objectives, with applications in two-sample tests and classification. This approach significantly enhances the performance of both two-sample tests and classification accuracy. **(c)** Given that the weighted MMD statistic is a U-statistic, maximizing it over the weights using gradient-based iterative algorithms often demands significant computational resources and time. To mitigate this challenge, we employ the first-order Taylor expansion of the weighted MMD statistic as a new objective function, which accelerates the optimization process. Furthermore, we present the convergence of this accelerated method and the convergence rate of the associated solving algorithm.

## 2 METHODOLOGY

### 2.1 PRELIMINARY ON MMD

Let $\mathcal{X} \subseteq \mathbb{R}^p$ be a metric space, and $\mathbf{F}$ and $\mathbf{G}$ be two $p$-dimensional distributions supported on $\mathcal{X}$. The two-sample hypothesis testing problem of interest is

$$H_0 : \mathbf{F} = \mathbf{G} \quad \text{versus} \quad H_1 : \mathbf{F} \neq \mathbf{G}. \tag{1}$$

The MMD, based on embeddings of $\mathbf{F}$ and $\mathbf{G}$ into a reproducing kernel Hilbert space (RKHS) $\mathcal{H}$, is introduced in Gretton et al. (2006; 2012) to test the hypothesis (1). Let $k(\cdot, \cdot) : \mathcal{X} \times \mathcal{X} \to \mathbb{R}$ be a kernel of $\mathcal{H}$ with feature map $k(\cdot, \mathbf{x}) \in \mathcal{H}$. Given independent random vectors $\boldsymbol{x}$ and $\boldsymbol{x}'$ from $\mathbf{F}$, and $\boldsymbol{y}$ and $\boldsymbol{y}'$ independently from $\mathbf{G}$, the squared population MMD under the conditions that $k(\cdot, \cdot)$ is measurable, $\mathbb{E}[k(\boldsymbol{x}, \boldsymbol{x})]^{1/2} < \infty$, and $\mathbb{E}[k(\boldsymbol{y}, \boldsymbol{y})]^{1/2} < \infty$ is defined as follows:

$$\mathrm{MMD}_k^2 = \mathbb{E}_{\boldsymbol{x}, \boldsymbol{x}'} [k(\boldsymbol{x}, \boldsymbol{x}')] - 2\mathbb{E}_{\boldsymbol{x}, \boldsymbol{y}} [k(\boldsymbol{x}, \boldsymbol{y})] + \mathbb{E}_{\boldsymbol{y}, \boldsymbol{y}'} [k(\boldsymbol{y}, \boldsymbol{y}')].$$

When $k$ is characteristic, $\mathrm{MMD}_k = 0$ if and only if $\mathbf{F} = \mathbf{Q}$. Many commonly used kernels, such as Gaussian and Laplace kernels, are characteristic (Fukumizu et al., 2007). Therefore, the MMD serves as an effective measure to quantify the differences between two multivariate distributions.

Suppose we observe independent i.i.d. samples $\mathfrak{X}_n = \{X_1, \ldots, X_n\} \sim \mathbf{F}$ and $\mathfrak{Y}_m = \{Y_1, \ldots, Y_m\} \sim \mathbf{G}$, an unbiased estimator of $\mathrm{MMD}_k^2$ is given by

$$\widehat{\mathrm{MMD}}_k^2 = \frac{1}{n(n-1)} \sum_{1 \leq i_1 \neq i_2 \leq n} k\left(X_{i_1}, X_{i_2}\right) + \frac{1}{m(m-1)} \sum_{1 \leq j_1 \neq j_2 \leq m} k\left(Y_{j_1}, Y_{j_2}\right)$$
$$- \frac{2}{nm} \sum_{i=1}^n \sum_{j=1}^m k\left(X_i, Y_j\right),$$

which is referred to as the empirical squared MMD. The null hypothesis $H_0 : \mathbf{F} = \mathbf{G}$ is rejected when $\widehat{\mathrm{MMD}}_k^2$ exceeds a critical value. The implementation of the MMD test typically employs a permutation test (Sutherland et al., 2017).

## 2.2 WEIGHTED MMD

When the null hypothesis in equation 1 is rejected, indicating a significant difference between the two distributions $\mathbf{F}$ and $\mathbf{G}$, a natural and important follow-up question is determining where these differences occur. In sparse scenarios (Tibshirani, 1996), this reduces to identifying the variables that contribute to the distributional discrepancy, which often involves a small subset of the total dimensions. In addition, inspired by the fact that variable selection is often a preliminary task for other objectives, in this paper, we consider the case where variable selection is guided by specific objects or tasks, a problem we refer to as object-oriented variable selection.

One common scenario occurs in the two-sample testing problem. It is discovered in Ramdas et al. (2015b) that the power of the MMD test diminishes as the number of non-informative variables increases but the signal remains constant. A promising solution is to identify a subset of variables for which the MMD test achieves optimal power (Mueller & Jaakkola, 2015). Similarly, in classification tasks, the goal is to select the subset of variables that maximizes classification accuracy, as the inclusion of non-important variables can degrade the performance of many classifiers (Andrews & McNicholas, 2013; Chen & Lee, 2020).

In the original versions of the MMD test and many distance-based classifiers (e.g., $k$-nearest neighbors and support vector machines), all variables are treated equally, with the same weight. However, assigning too much weight to non-important variables can lead to issues such as reduced test power and decreased classification accuracy. By contrast, if we assign higher weights to important variables and near-zero weights to non-informative ones, the negative impact of non-important variables can be mitigated, potentially leading to increased test power and improved classification accuracy.

Figure 1 illustrates the impact of variable weighting on the power of the MMD test and the classification accuracy of a feedforward neural network with one hidden layer. In this toy example, the data are drawn from $\mathbf{X} \sim N(\mu_1, \Sigma_1)$ and $\mathbf{Y} \sim N(\mu_2, \sigma_2)$ with $\mu_1, \mu_2, \Sigma_1$ and $\Sigma$ specified under different scenarios. The results clearly show that increasing the weights for the important variables (the first three dimensions) while simultaneously decreasing the weights for other variables leads to improvements in both MMD test power and classification accuracy. However, the optimal weighting scheme may vary across different scenarios, even when aiming for the same target.

Motivated by the above discussion, we propose the weighted MMD, which serves as the foundation for the regularized MMD. In this paper, we consider the isotropic kernel of the form $k\left(x, y\right) = f(\|x - y\|_2^2/\gamma)$, where $f(\cdot)$ is a real-valued function on $[0, \infty)$ and $\gamma > 0$ is a bandwidth parameter. This framework includes commonly used kernels such as the Gaussian kernel $f(x) = \exp(-x)$, the Laplacian kernel $f(x) = \exp(-\sqrt{x})$, the rational quadratic kernel $f(x) = (1+x)^{-\alpha}$ for $\alpha > 0$, and so on (Yan & Zhang, 2023). For two random vectors $x = (x_1, \ldots, x_p)^\top$ and $y = (y_1, \ldots, y_p)^\top$, their Euclidean distance is defined as $\|x - y\|_2 = \{\sum_{r=1}^p (x_r - y_r)^2\}^{1/2}$. Introducing a weight vector $\mathbf{w} = (w_1, \ldots, w_p)^\top$, we define the weighted Euclidean distance between $x$ and $y$ as

$$\|x - y\|_{\mathbf{w}} = \left\{\sum_{r=1}^p w_r (x_r - y_r)^2\right\}^{1/2}.$$

Substituting the weighted distance into MMD, we obtain the weighted population MMD:

$$\mathrm{MMD}_{f,\mathbf{w}}^2 = \mathbb{E}_{x,x'}\left[f(\|x - x'\|_{\mathbf{w}}^2/\gamma)\right] - 2\mathbb{E}_{x,y}\left[f(\|x - y\|_{\mathbf{w}}^2/\gamma)\right] + \mathbb{E}_{y,y'}\left[f(\|y - y'\|_{\mathbf{w}}^2/\gamma)\right].$$

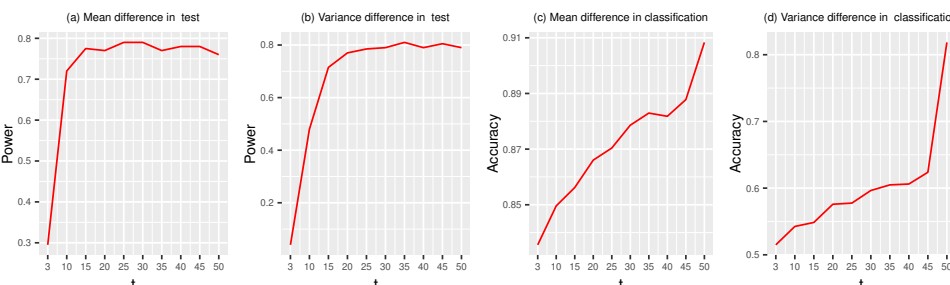

Figure 1: The samples $\mathfrak{X}_n$ and $\mathfrak{Y}_m$ are drawn from $\mathbf{X} \sim N(\mu_1, \Sigma_1)$, $\mathbf{Y} \sim N(\mu_2, \Sigma_2)$, with $n = m = 100$ and $p = 50$. (a): $\mu_1 = (\mathbf{0.3}_3^\top, \mathbf{0}_{47}^\top)^\top$, $\mu_2 = \mathbf{0}_{50}$, $\Sigma_1 = \Sigma_2 = \{(0.2)^{|i-j|}\}_{1 \le i,j \le 50}$; (b): $\mu_1 = \mu_2 = \mathbf{0}_{50}$, $\Sigma_1 = \{(0.2)^{|i-j|}\}_{1 \le i,j \le 50}$, $\Sigma_2 = \mathrm{diag}(\mathbf{1}_3, \mathbf{0}_{47}) + \Sigma_1$; (c): $\mu_1 = (\mathbf{4}_3^\top, \mathbf{0}_{47}\top)^\top$, $\mu_2 = \mathbf{0}_{50}$, $\Sigma_1 = \Sigma_2 = \{(0.2)^{|i-j|}\}_{1 \le i,j \le 50}$; (d): $\mu_1 = \mu_2 = \mathbf{0}_{50}$, $\Sigma_1 = \{(0.2)^{|i-j|}\}_{1 \le i,j \le 50}$, $\Sigma_2 = \mathrm{diag}(\mathbf{4}_3, \mathbf{0}_{47}) + \Sigma_1$. The curves show empirical power in (a) and (b) or classification accuracy in (c) and (d) based on $(\mathbf{w}_t^{1/2})^\top \mathfrak{X}_n$, and $(\mathbf{w}_t^{1/2})^\top \mathfrak{Y}_n$, where $\mathbf{w}_t = ((t/3)\mathbf{1}_3^\top, ((50-t)/47)\mathbf{1}_{47}^\top)^\top$.

From another perspective, $\mathrm{MMD}_{f,\mathbf{w}}^2$ can be viewed as the MMD applied to $\mathbf{w}^{1/2}\mathbf{X}$ and $\mathbf{w}^{1/2}\mathbf{Y}$.

Given $\mathfrak{X}_n$ and $\mathfrak{Y}_m$, the empirical weighted squared MMD is defined as

$$\widehat{\mathrm{MMD}}_{f,\mathbf{w}}^2 = \frac{1}{n(n-1)} \sum_{1 \le i_1 \ne i_2 \le n} f(\|\boldsymbol{X}_{i_1} - \boldsymbol{X}_{i_2}\|_{\mathbf{w}}^2 / \gamma) - \frac{2}{nm} \sum_{i=1}^{n} \sum_{j=1}^{m} f(\|\boldsymbol{X}_i - \boldsymbol{Y}_j\|_{\mathbf{w}}^2 / \gamma)$$

$$+ \frac{1}{m(m-1)} \sum_{1 \le j_1 \ne j_2 \le m} f(\|\boldsymbol{Y}_{j_1} - \boldsymbol{Y}_{j_2}\|_{\mathbf{w}}^2 / \gamma).$$

which serves as an unbiased estimator of $\mathrm{MMD}_{f,\mathbf{w}}^2$.

We now turn to identifying important variables using the assigned weights. Intuitively, increasing the weight of important variables should result in a significant rise in the MMD, while increasing the weight of non-important variables should have the opposite effect, diminishing the MMD. This observation motivates us to maximize the weighted MMD with respect to the assigned weights.

## 2.3 REGULARIZATION ON WEIGHTED MMD

To obtain the optimal weight vector that maximizes $\mathrm{MMD}_{f,\mathbf{w}}^2$, it is crucial to address the challenges posed by significant differences in signal strength among important variables. When these differences exist, the process of maximizing the MMD tends to favor variables with higher signal strength, resulting in an imbalance in weight allocation. Specifically, high-signal-strength variables receive disproportionately large weights, while low-signal-strength variables are often overlooked. This dynamic can lead to the weights of less influential variables approaching zero, effectively rendering them unselectable. To counter this bias, we propose incorporating a regularization term into the objective function. This term mitigates the risk of excessively large weights for high-signal-strength variables.

In this paper, we utilize $\ell_2$-regularization (ridge penalty) for its computational flexibility, although other types of regularization could also be employed. Additionally, the weights are further constrained to be non-negative and sum to $p$, as required under undisturbed conditions. In summary, the optimal weights $\mathbf{w}^*$ are obtained by solving the following constrained minimization problem:

$$\text{minimize } -\mathrm{MMD}_{f,\mathbf{w}}^2 + \lambda \sum_{r=1}^{p} \omega_r^2, \text{ subject to } \sum_{r=1}^{p} \omega_r = p \text{ and } \omega_r \ge 0 \text{ for } r = 1, \dots, p. \quad (2)$$

where $\lambda > 0$ is a tuning parameter. Let $\boldsymbol{\Omega} = \{\mathbf{w} : \sum_{r=1}^{p} \omega_r = p \text{ and } \omega_r \ge 0 \text{ for } r = 1, \dots, p\}$, the optimization problem in equation 2 can then be rewritten as:

$$\mathbf{w}_\lambda^* = \underset{\mathbf{w} \in \boldsymbol{\Omega}}{\mathrm{argmin}} \ -\mathrm{MMD}_{f,\mathbf{w}}^2 + \lambda \sum_{r=1}^{p} \omega_r^2.$$

Suppose we observe independent random samples $\mathfrak{X}_n$ and $\mathfrak{Y}_m$, the estimator of the optimal weight vector $\mathbf{w}_\lambda^*$ is given by

$$\widehat{\mathbf{w}}_\lambda = \operatorname*{argmin}_{\mathbf{w} \in \mathbf{\Omega}} \; -\widehat{\mathrm{MMD}}_{f,\mathbf{w}}^2 + \lambda \sum_{r=1}^{p} \omega_r^2. \tag{3}$$

The optimal weights $\widehat{\mathbf{w}}_\lambda$ serve as important measures of variables that contribute to the differences between two distributions. Variables with significantly positive weights, or weights exceeding a pre-specified positive threshold, are considered important and selected. In contrast, variables with weights close to zero are regarded as non-important.

When $f(\cdot)$ is a convex function, which is true for most kernel functions, $\widehat{\mathbf{w}}_\lambda$ can be obtained by combining the difference of convex functions algorithm (Thi & Dinh, 2018) with the mirror descent algorithm (Amir, 2017) to solve equation 3, as detailed in Algorithm 3 in Appendix A.1.2. Additionally, in the Appendix A.2, we present the impact of the presence or absence of a regularization term on variable selection under scenarios with significant differences in the signal strength of important variables.

## 2.4 OBJECT-ORIENTED VARIABLE SELECTION

Given that the optimization problem in equation 3 is equivalent to

$$\widehat{\mathbf{w}}_\lambda = \operatorname*{argmin}_{\mathbf{w} \in \mathbf{\Omega}} \; -\widehat{\mathrm{MMD}}_{f,\mathbf{w}}^2 + \lambda \sum_{r=1}^{p} (\omega_r - 1)^2,$$

the tuning parameter $\lambda$ plays a crucial role as a regularization term that controls the number of important variables. Specifically, it controls how much the weights are allowed to deviate from 1. A larger $\lambda$ penalizes deviations from 1 more heavily, promoting a more uniform set of weights and encouraging the inclusion of more variables. In contrast, a smaller $\lambda$ allows greater flexibility in the weights, resulting in more extreme values where only the most significant variables are selected, leading to a sparser model.

In practice, many variable selection problems serve as a guideline for subsequent tasks. For instance, variable selection is commonly used to reduce the dimensionality of data in classification, thereby addressing the challenges posed by high-dimensionality. In such cases, the primary objective is to optimize classification accuracy. To this end, we propose an object-oriented variable selection approach, wherein the tuning parameter is determined by optimizing a related task aligned with specific objectives. Let $\ell(\lambda)$ represent the loss function of the objective of interest, which is influenced by the variable selection based on the regularized MMD and, consequently, depends on $\lambda$. The tuning parameter is then selected as the value of $\lambda$ that minimizes $\ell(\lambda)$. In summary, the object-oriented variable selection approach based on the regularized MMD can be expressed as follows:

$$\widehat{\mathbf{w}}_{\hat\lambda} = \operatorname*{argmin}_{\mathbf{w} \in \mathbf{\Omega}} \; -\widehat{\mathrm{MMD}}_{f,\mathbf{w}}^2 + \hat\lambda \sum_{r=1}^{p} \omega_r^2, \quad \text{where} \;\; \hat\lambda = \operatorname*{argmin}_{\lambda > 0} \ell(\lambda). \tag{4}$$

It is generally required that the data used for variable selection be independent of the data used for the final objective (e.g., a two-sample test or classification). This can be achieved through data splitting or cross-validation, ensuring that the selection process does not bias the final analysis.

We now illustrate the implementation of the proposed approach using two common applications: the two-sample test and binary classification. In the two-sample test problem, it is reasonable to choose the power function as $-\ell(\lambda)$. The optimal tuning parameter $\hat\lambda$ is then determined by maximizing this power function. As suggested by Sutherland et al. (2017); Liu et al. (2020), the loss function $\ell(\lambda)$ can be expressed as:

$$\ell(\lambda) = -\widehat{\mathrm{MMD}}_{f,\widehat{\mathbf{w}}_\lambda}^2 / \hat\sigma_{f,\widehat{\mathbf{w}}_\lambda}, \tag{5}$$

which is the main contribution term in the power function. Here, $\widehat{\mathrm{MMD}}_{f,\widehat{\mathbf{w}}_\lambda}^2$ is the empirical squared MMD computed using the weight vector $\widehat{\mathbf{w}}_\lambda$, and $\hat\sigma_{f,\widehat{\mathbf{w}}_\lambda}^2$ is an estimator of its variance (Sutherland et al., 2017; Sutherland & Deka, 2022), which is detailed in Appendix A.3. To implement the MMD test following object-oriented variable selection, the dataset needs to be split into two disjoint

subsets. The first subset, the training set, is used to compute the optimal weight vector $\widehat{\mathbf{w}}_{\hat{\lambda}}$ by optimizing the tuning parameter $\hat{\lambda}$. The second subset, the test set, is used to perform the permutation MMD test, utilizing the weighted MMD with the computed weight vector $\widehat{\mathbf{w}}_{\hat{\lambda}}$. The complete process is outlined in Algorithm 1.

---

**Algorithm 1** Regularized-MMD variable selection for two-sample test

---

**Require:**
   Samples $\mathfrak{X}_n, \mathfrak{Y}_m$, a sequence of tuning parameters $\lambda_1, \cdots, \lambda_q$;
   Split data as $\mathfrak{X} = \mathfrak{X}^{tr} \cup \mathfrak{X}^{te}$ and $\mathfrak{Y} = \mathfrak{Y}^{tr} \cup \mathfrak{Y}^{te}$.
   #Phase 1: select the optimal tuning parameter.
   **For** $j = 1, \ldots, q$ **do**
      Select $\lambda_j$, obtained $\widehat{\mathbf{w}}_{\lambda_j}$ by solving equation 3.
      Compute $\ell(\lambda_j) = -\widehat{\mathrm{MMD}}^2_{f,\widehat{\mathbf{w}}_{\lambda_j}}/\hat{\sigma}_{f,\widehat{\mathbf{w}}_{\lambda_j}}$.
   **End for**
   Let $\hat{\lambda} = \underset{\lambda_j, j=1,\cdots,q}{\operatorname{argmin}} \ell(\lambda_j)$, $\mathfrak{X}^{te}_{\widehat{\mathbf{w}}_{\hat{\lambda}}}$ and $\mathfrak{Y}^{te}_{\widehat{\mathbf{w}}_{\hat{\lambda}}}$ represents the test samples that contain only selected
   important variables by $\widehat{\mathbf{w}}_{\hat{\lambda}}$.
   #Phase 2: permutation test based on $\mathfrak{X}^{te}_{\widehat{\mathbf{w}}_{\hat{\lambda}}}, \mathfrak{Y}^{te}_{\widehat{\mathbf{w}}_{\hat{\lambda}}}$.
   Compute $M_p = \widehat{\mathrm{MMD}}^2_f$, based on $\mathfrak{X}^{te}_{\widehat{\mathbf{w}}_{\hat{\lambda}}}$ and $\mathfrak{Y}^{te}_{\widehat{\mathbf{w}}_{\hat{\lambda}}}$.
   Randomly partition $\mathfrak{X}^{te}_{\widehat{\mathbf{w}}_{\hat{\lambda}}} \cup \mathfrak{Y}^{te}_{\widehat{\mathbf{w}}_{\lambda}}$ into $\mathfrak{X}^i_{\widehat{\mathbf{w}}_{\hat{\lambda}}}$ and $\mathfrak{Y}^i_{\widehat{\mathbf{w}}_{\hat{\lambda}}}$ $n_p$ times.
   Compute $M_i = \widehat{\mathrm{MMD}}^2_f$, based on $\mathfrak{X}^i_{\widehat{\mathbf{w}}_{\hat{\lambda}}}$ and $\mathfrak{Y}^i_{\widehat{\mathbf{w}}_{\hat{\lambda}}}$, $i = 1, \cdots, n_p$.
**Ensure:**
   $\widehat{\mathbf{w}}_{\hat{\lambda}}, M_p$, p-value: $n_p^{-1} \sum_{i=1}^{n_p} \mathbb{1}(M_i > M_p)$.

---

In binary classification, the loss function $\ell(\lambda)$ is chosen as the misclassification error, and the optimal tuning parameter $\hat{\lambda}$ is selected through cross-validation. The detailed procedure for classification with object-oriented variable selection is outlined in Algorithm 2 in Appendix A.1.1. This approach is highly flexible and can be seamlessly integrated with any classification algorithm.

## 2.5 ACCELERATED COMPUTATION METHOD

Denote $h_{f,\mathbf{w}}(\boldsymbol{x}, \boldsymbol{x}'; \boldsymbol{y}, \boldsymbol{y}') = f(\|\boldsymbol{x} - \boldsymbol{x}'\|^2_{\mathbf{w}}/\gamma) + f(\|\boldsymbol{y} - \boldsymbol{y}'\|^2_{\mathbf{w}}/\gamma) - f(\|\boldsymbol{x} - \boldsymbol{y}'\|^2_{\mathbf{w}}/\gamma) - f(\|\boldsymbol{x}' - \boldsymbol{y}\|^2_{\mathbf{w}}/\gamma)$. The empirical weighted MMD, $\widehat{\mathrm{MMD}}^2_{f,\mathbf{w}}$, is a two-sample $U$-statistic with kernel function $h_{f,\mathbf{w}}(\boldsymbol{x}, \boldsymbol{x}'; \boldsymbol{y}, \boldsymbol{y}')$ of degree $(2, 2)$, and its computational complexity is of order $O(n^2 m^2)$ (Huang et al., 2023). This makes it computationally intensive for large sample sizes. Moreover, optimizing equation 3 involves computing $p$ statistics of the same order of complexity as $\widehat{\mathrm{MMD}}^2_{f,\mathbf{w}}$ at each iteration, further escalating the computational cost. To mitigate this issue, we propose an accelerated algorithm by applying a first-order Taylor expansion of $\widehat{\mathrm{MMD}}^2_{f,\mathbf{w}}$ around $\mathbf{w} = \mathbf{1}_p$, transforming the objective function into a linear form that simplifies optimization and reduces computation time.

First, we present the first derivative of $\mathrm{MMD}^2_{f,\mathbf{w}}$ at $\mathbf{w} = \mathbf{1}_p$. Assume that $f(\cdot)$ is differentiable, and let $f^{(1)}(\cdot)$ be its first derivative. Define $\mathrm{df}_r(\boldsymbol{x}, \boldsymbol{y}) = \gamma^{-1}(x_r - y_r)^2 f^{(1)}(\|\boldsymbol{x} - \boldsymbol{y}\|^2/\gamma)$ for $r = 1, \ldots, p$. Assuming the exchangeability of differentiation and integration, the derivative of $\mathrm{MMD}^2_{f,\mathbf{w}}$ with respect to $w_r$, evaluated at $\mathbf{w} = \mathbf{1}_p$, is given by

$$\mathrm{dMMD}^{2(r)}_f = \frac{\partial \mathrm{MMD}^2_{f,\mathbf{w}}}{\partial w_r}\bigg|_{\mathbf{w}=\mathbf{1}_p} = \mathbb{E}_{\boldsymbol{x},\boldsymbol{x}'}[\mathrm{df}_r(\boldsymbol{x}, \boldsymbol{x}')] - 2\mathbb{E}_{\boldsymbol{x},\boldsymbol{y}'}[\mathrm{df}_r(\boldsymbol{x}, \boldsymbol{y}')] + \mathbb{E}_{\boldsymbol{y},\boldsymbol{y}'}[\mathrm{df}_r(\boldsymbol{y}, \boldsymbol{y}')],$$

for $r = 1, \cdots, p$. We can then perform a first-order Taylor expansion of $\mathrm{MMD}^2_{f,\mathbf{w}}$ as follows

$$\mathrm{MMD}^2_{f,\mathbf{w}} = \mathrm{MMD}^2_f + \left[\mathrm{dMMD}^2_f\right]^\top (\mathbf{w} - \mathbf{1}_p) + O(\|\mathbf{w} - \mathbf{1}_p\|^2),$$

where $\mathrm{dMMD}_f^2 = (\mathrm{dMMD}_f^{2(1)}, \cdots, \mathrm{dMMD}_f^{2(p)})^\top$. Using this expansion, we propose the accelerated version of the optimal weight vector $\mathbf{w}_\lambda^*$, solving the following optimization problem:

$$\mathbf{w}_\lambda^* = \underset{\mathbf{w} \in \boldsymbol{\Omega}}{\mathrm{argmin}} \;\; -[\mathrm{dMMD}_f^2]^\top \mathbf{w} + \lambda \sum_{r=1}^p \omega_r^2.$$

Given $\mathfrak{X}_n$ and $\mathfrak{Y}_m$, the estimator of the accelerated optimal weight vector is defined as

$$\widehat{\mathbf{w}}_\lambda = \underset{\mathbf{w} \in \boldsymbol{\Omega}}{\mathrm{argmin}} \; - \left[\widehat{\mathrm{dMMD}}_f^2\right]^\top \mathbf{w} + \lambda \sum_{r=1}^p \omega_r^2. \tag{6}$$

where $\widehat{\mathrm{dMMD}}_f^2 = (\widehat{\mathrm{dMMD}}_f^{2(1)}, \cdots, \widehat{\mathrm{dMMD}}_f^{2(p)})^\top$ and

$$\widehat{\mathrm{dMMD}}_f^{2(r)} = \frac{1}{n(n-1)} \sum_{1 \le i_1 \ne i_2 \le n} \mathrm{df}_r(\boldsymbol{X}_{i_1}, \boldsymbol{X}_{i_2}) - \frac{2}{nm} \sum_{i=1}^n \sum_{j=1}^m \mathrm{df}_r(\boldsymbol{X}_i, \boldsymbol{Y}_j)$$

$$+ \frac{1}{m(m-1)} \sum_{1 \le j_1 \ne j_2 \le m} \mathrm{df}_r(\boldsymbol{Y}_{j_1}, \boldsymbol{Y}_{j_2}).$$

To solve this optimization problem, the mirror gradient descent algorithm can be applied. The detailed algorithm is provided in Algorithm 4 in Appendix A.1.2. In Algorithm 3, all partial derivatives of $f(\|\boldsymbol{x} - \boldsymbol{x}'\|_{\mathbf{w}}^2/\gamma)$ need to be computed in each iteration. Therefore, given the iteration step size $T$ and $T_1$ in Algorithm 3, its computational complexity is $O(TT_1 pnm) + O(Tp(n^2 + m^2))$. In contrast, the accelerated algorithm only needs to compute all partial derivatives of $f(\|\boldsymbol{x} - \boldsymbol{x}'\|_{\mathbf{w}}^2/\gamma)$ once, so its computational complexity is $O(p(nm + n^2 + m^2))$. Thus, the accelerated algorithm is significantly faster than the original method.

Table 1 presents the computation times for obtaining the optimal weights through both the original method based on equation 3 and the accelerated method derived from equation 6. The data were from $\mathbf{X} \sim N(\mu_1, \Sigma)$ and $\mathbf{Y} \sim N(\mu_2, \Sigma)$, where $\mu_1 = ((4, 3, 2, 1)^\top, \mathbf{0}_{p-4}^\top)^\top$, $\mu_2 = \mathbf{0}_p$, $\Sigma = \{(0.2)^{|i-j|}\}_{1 \le i,j \le p}$ with $n = m = 200$. The results demonstrate that the accelerated algorithm substantially reduces computation time compared to the original algorithm, particularly as the dimensionality $p$ increases. Furthermore, Appendix A.4 provides a comparative analysis of the results obtained from both methods in the context of binary classification, highlighting that the accelerated method maintains satisfactory performance while achieving significant savings in computation time.

Table 1: Computation times of the original method and the accelerated method (in seconds).

| $p$ | 5 | 50 | 100 | 150 |
|---|---|---|---|---|
| Original | 4.61 | 11.7 | 53.92 | 82.4 |
| Accelerated | 0.06 | 0.11 | 0.23 | 0.31 |

## 3 THEORETICAL PROPERTIES

In this section, we establish the theoretical properties of the optimal weight vector estimator $\widehat{\mathbf{w}}_\lambda$ obtained from the accelerated method. Specifically, we demonstrate the consistency of $\widehat{\mathbf{w}}_\lambda$ and the convergence of the accelerated mirror gradient descent algorithm.

**Assumption 1.** *Let $N = n + m$, there exists a constant $0 < c_1 < 1$ such that $\lim_{N \to \infty} n/N = c_1$.*

**Assumption 2.** *$f(\cdot)$ is differentiable with a first derivative $f^{(1)}(\cdot)$ that satisfies $\sup_{x \ge 0} |f^{(1)}(x)| \le B_1$ for some positive constant $B_1$.*

**Assumption 3.** *There exists a positive constant $s > 4$ such that*

$$\sup_p \max_{1 \le r \le p} \mathbb{E}\left(|X_{1,r}|^{2s}\right) < \infty \quad \text{and} \quad \sup_p \max_{1 \le r \le p} \mathbb{E}\left(|Y_{1,r}|^{2s}\right) < \infty.$$

Assumption 1 ensures that the sample sizes $n$ and $m$ are of the same order. For the kernel of the form $f(\|\boldsymbol{x} - \boldsymbol{y}\|_2^2/\gamma)$, Assumption 2 is satisfied by Gaussian kernel, Laplacian kernel, and so on. Assumption 3 is a moderate moment condition that is satisfied by a wide range of distributions.

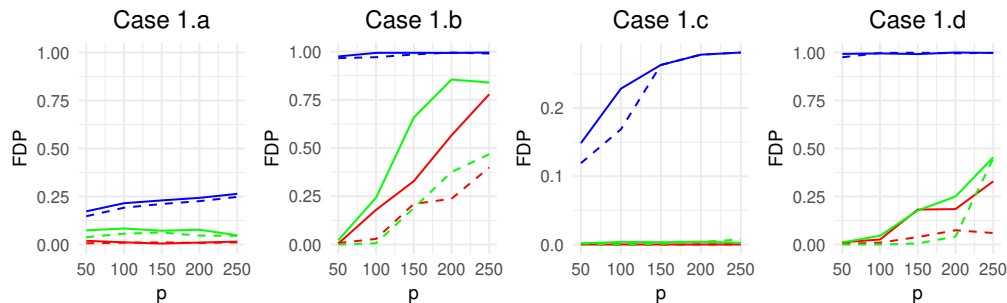

Figure 2: The averages FDP (y-axis) over 250 replications across different dimensions (x-axis) in binary classification. The solid curves correspond to cases with 4 important variables, and the dashed curves represent cases with 8 important variables. The red curves represent rMMD-c, and the green curves represent rMMD-c, while the blue lines depict the n-Lasso method.

**Theorem 1.** *Under Assumptions $1 - 3$, let $\widehat{\mathbf{w}}_\lambda$ be the accelerated optimal weight vector obtained from equation 6. Then as $N \to \infty$, we have*

$$\widehat{\mathbf{w}}_\lambda \xrightarrow{P} \mathbf{w}_\lambda^*.$$

**Theorem 2.** *When applying the mirror descent algorithm to solve equation 6, the optimal step size for the iterative process at step $T$ is given by*

$$\alpha_T = \sqrt{\frac{2}{T+1}} L_f^{-1}, \quad where \quad L_f = \max_{1 \leq r \leq p} \left( \left| -\widehat{\mathrm{dMMD}}_f^{2(r)} \right|, \left| -\widehat{\mathrm{dMMD}}_f^{2(r)} + 2p\lambda \right| \right).$$

*Moreover, let $Z_N(w) = -[\mathrm{dMMD}_f^2]^\top \mathbf{w} + \lambda \sum_{r=1}^p \omega_r^2$, and $\mathbf{w}^t$ be the solution at the $t$-th iteration. The convergence rate of the mirror descent algorithm, using the optimal step size and under the sample space measured by the $\ell_1$-norm, is given by:*

$$\min_{t=0,\dots,T} Z_N(\mathbf{w}^t) - \min_{\mathbf{w} \in \mathbf{\Omega}} Z_N(\mathbf{w}) \leq \sqrt{\frac{2}{T+1}} L_f.$$

The optimal convergence rate indicates that after $T$ iterations, $\min_{t=0,\dots,T} Z_N(\mathbf{w}^t)$ will converge to the global minimum at the rate of $O(1/\sqrt{T})$.

## 4 NUMERICAL EXPERMENTS

**Methods**. In two-sample tests, we employ Algorithm 1 (rMMD-t), which uses the accelerated computation method for variable selection, and compare it to the original MMD test (MMD) (Gretton et al., 2012), which uses all variables. In this section, we present the results under the alternative hypothesis holding. The results under the null hypothesis holding can be found in the Appendix A.5.3. For binary classification, we use Algorithm 2 (rMMD-c) with a feedforward neural network (FNN) classifier, also based on the accelerated computation method for variable selection. This approach is compared to an FNN model with variables selected by Lasso (Hastie et al., 2009) (n-Lasso) and an FNN model trained on all available variables (all). In addiction, Algorithm 2 (rMMD-k) with a k-nearest neighbors (k-NN) classifier is also considered.

**Synthetic Data 1**. In the first set of synthetic data, the two samples differ only in their marginal distributions. We consider six different cases for both binary classification and two-sample tests, with detailed settings provided in Appendix A.5.1. Table 2 presents the classification accuracy across different scenarios, and the standard deviation is shown in parentheses. And test power is shown in Table 7. Figure 2 plot the False Discovery Proportion (FDP) for variable selection in binary classification under Cases 1.a–1.d. If no variables are selected, an FDP value of 1 is assigned. The FDP results for the remaining two cases are reported in Appendix A.5.4.

**Synthetic Data 2.** We explore a different scenario in the second set of synthetic data, where the differences occur in the joint distributions rather than the marginals. The detailed settings for

Table 2: Mean of classification accuracy in Synthetic Data 1.

| method\ dimension | 50 | 100 | 150 | 200 | 250 |
|---|---|---|---|---|---|
| Case 1.a, $\mathbf{I}^* = 4$ | | | | | |
| rMMD-c | 0.705 (0.035) | 0.703 (0.038) | 0.709 (0.034) | 0.709 (0.037) | **0.709** (0.035) |
| rMMD-k | **0.753** (0.036) | **0.750** (0.034) | **0.752** (0.032) | **0.751** (0.031) | 0.702 (0.034) |
| n-Lasso | 0.686 (0.035) | 0.688 (0.036) | 0.685 (0.036) | 0.686 (0.035) | 0.683 (0.037) |
| all | 0.604 (0.044) | 0.595 (0.031) | 0.589 (0.026) | 0.589 (0.021) | 0.584 (0.039) |
| Case 1.b, $\mathbf{I}^* = 4$ | | | | | |
| rMMD-c | 0.708 (0.048) | **0.722** (0.056) | **0.688** (0.069) | **0.644** (0.071) | 0.618 (0.082) |
| rMMD-k | **0.737** (0.033) | 0.720 (0.043) | 0.682 (0.042) | 0.634 (0.040) | **0.627** (0.045) |
| n-Lasso | 0.508 (0.033) | 0.502 (0.035) | 0.503 (0.036) | 0.503 (0.032) | 0.503 (0.034) |
| all | 0.500 (0.033) | 0.500 (0.035) | 0.499 (0.033) | 0.498 (0.031) | 0.500 (0.036) |
| Case 1.c, $\mathbf{I}^* = 4$ | | | | | |
| rMMD-c | **0.914** (0.041) | **0.914** (0.046) | **0.912** (0.047) | **0.914** (0.051) | **0.913** (0.060) |
| rMMD-k | 0.911 (0.021) | 0.912 (0.020) | 0.910 (0.018) | 0.911 (0.021) | 0.909 (0.020) |
| n-Lasso | 0.903 (0.042) | 0.896 (0.048) | 0.897 (0.050) | 0.891 (0.053) | 0.891 (0.061) |
| all | 0.848 (0.044) | 0.820 (0.031) | 0.797 (0.026) | 0.772 (0.021) | 0.760 (0.016) |
| Case 1.d, $\mathbf{I}^* = 4$ | | | | | |
| rMMD-c | 0.740(0.028) | 0.740 (0.030) | 0.721 (0.034) | 0.652 (0.036) | 0.699 (0.041) |
| rMMD-k | **0.776** (0.028) | **0.774** (0.029) | **0.763** (0.031) | **0.721** (0.037) | **0.721** (0.042) |
| n-Lasso | 0.500 (0.033) | 0.508 (0.035) | 0.503 (0.033) | 0.502 (0.034) | 0.502 (0.035) |
| all | 0.537 (0.034) | 0.514 (0.036) | 0.506 (0.035) | 0.502 (0.037) | 0.501 (0.034) |
| Case 1.e, $\mathbf{I}^* = 4$ | | | | | |
| rMMD-c | 0.846 (0.033) | 0.848 (0.036) | 0.845 (0.038) | 0.830 (0.041) | 0.810 (0.047) |
| rMMD-k | **0.860** (0.026) | **0.860** (0.028) | **0.858** (0.030) | **0.842** (0.037) | **0.816** (0.051) |
| n-Lasso | 0.513 (0.032) | 0.509 (0.034) | 0.513 (0.033) | 0.505(0.035) | 0.513 (0.037) |
| all | 0.510 (0.031) | 0.520 (0.033) | 0.501 (0.029) | 0.500 (0.035) | 0.500 (0.033)) |
| Case 1.f, $\mathbf{I}^* = 4$ | | | | | |
| rMMD-c | **0.838** (0.063) | **0.701** (0.091) | **0.614** (0.084) | **0.576** (0.092) | **0.553** (0.087) |
| rMMD-k | 0.659 (0.092) | 0.544 (0.046) | 0.521 (0.032) | 0.514 (0.036) | 0.510 (0.035) |
| n-Lasso | 0.524 (0.038) | 0.506 (0.033) | 0.507 (0.035) | 0.508 (0.035) | 0.513 (0.034) |
| all | 0.633 (0.076) | 0.520 (0.054) | 0.506 (0.048) | 0.500 (0.041) | 0.503 (0.033) |
| Case 1.a, $\mathbf{I}^* = 8$ | | | | | |
| rMMD-c | 0.744 (0.033) | 0.741 (0.035) | 0.744 (0.037) | 0.742 (0.034) | 0.743 (0.033) |
| rMMD-k | **0.752** (0.031) | **0.750** (0.034) | **0.754** (0.040) | **0.753** (0.041) | **0.754** (0.048) |
| n-Lasso | 0.729 (0.034) | 0.715 (0.035) | 0.723 (0.036) | 0.715 (0.033) | 0.718 (0.034) |
| all | 0.652 (0.040) | 0.637 (0.041) | 0.638 (0.025) | 0.628 (0.019) | 0.625 (0.039) |
| Case 1.b, $\mathbf{I}^* = 8$ | | | | | |
| rMMD-c | **0.753** (0.032) | **0.749** (0.037) | **0.733** (0.056) | **0.712** (0.051) | **0.690** (0.066) |
| rMMD-k | 0.746 (0.031) | 0.742 (0.034) | 0.719 (0.040) | 0.654 (0.041) | 0.690 (0.048) |
| n-Lasso | 0.514 (0.033) | 0.506 (0.033) | 0.504 (0.035) | 0.503 (0.033) | 0.503 (0.034) |
| all | 0.499 (0.032) | 0.499 (0.036) | 0.500 (0.034) | 0.498 (0.033) | 0.500 (0.035) |
| Case 1.c, $\mathbf{I}^* = 8$ | | | | | |
| rMMD-c | 0.942 (0.045) | **0.943** (0.047) | 0.943 (0.051) | 0.944 (0.050) | 0.944 (0.047) |
| rMMD-k | **0.948** (0.014) | 0.927 (0.022) | **0.949** (0.015) | **0.950** (0.015) | **0.951** (0.015) |
| n-Lasso | 0.937 (0.050) | 0.937 (0.049) | 0.933 (0.052) | 0.934 (0.056) | 0.933 (0.059) |
| all | 0.903 (0.051) | 0.882 (0.041) | 0.858 (0.025) | 0.849 (0.019) | 0.826 (0.017) |
| Case 1.d, $\mathbf{I}^* = 8$ | | | | | |
| rMMD-c | 0.728 (0.033) | 0.733 (0.037) | 0.720 (0.035) | 0.718 (0.037) | **0.722** (0.039) |
| rMMD-k | **0.773** (0.036) | **0.746** (0.038) | **0.733** (0.035) | **0.721** (0.035) | 0.720 (0.037) |
| n-Lasso | 0.521 (0.036) | 0.514 (0.035) | 0.508 (0.033) | 0.507 (0.034) | 0.502 (0.036) |
| all | 0.565 (0.033) | 0.528 (0.037) | 0.513 (0.037) | 0.507 (0.039) | 0.502 (0.038) |
| Case 1.e, $\mathbf{I}^* = 8$ | | | | | |
| rMMD-c | 0.819 (0.035) | 0.819 (0.039) | 0.824 (0.042) | 0.827 (0.044) | **0.825** (0.048) |
| rMMD-k | **0.863** (0.035) | **0.861** (0.041) | **0.857** (0.042) | **0.844** (0.051) | 0.821 (0.046) |
| n-Lasso | 0.541 (0.036) | 0.532 (0.035) | 0.520 (0.032) | 0.515 (0.033) | 0.511 (0.037) |
| all | 0.521 (0.031) | 0.499 (0.032) | 0.499 (0.035) | 0.497 (0.033) | 0.496 (0.036) |
| Case 1.f, $\mathbf{I}^* = 8$ | | | | | |
| rMMD-c | **0.855** (0.062) | **0.777** (0.069) | **0.715** (0.084) | **0.642** (0.091) | **0.607** (0.099) |
| rMMD-k | 0.655 (0.093) | 0.584 (0.083) | 0.536 (0.053) | 0.523 (0.034) | 0.515 (0.030) |
| n-Lasso | 0.543 (0.040) | 0.526 (0.037) | 0.517 (0.035) | 0.513 (0.036) | 0.515 (0.035) |
| all | 0.665 (0.063) | 0.553 (0.055) | 0.513 (0.051) | 0.502 (0.042) | 0.501 (0.034) |

generating samples are outlined in Appendix A.5.2, and the results are presented in Table 3, and the standard deviations of classification accuracy are shown in Table 8.

**Gene expression dataset.** We utilize GSE2034 gene dataset from the Gene Expression Profiles of Breast Cancer study (Xie et al., 2017), which forms a binary classification problem. This dataset contains $12,634$ genes, and we preprocess it using Min-Max normalization. For the training set, we select the first 75 recurrence tumor samples and the first 100 non-recurrence samples. The

Table 3: Mean of classification accuracy and test power in Synthetic Data 2.

| I* | | 3 | | | | | 6 | | | |
|---|---|---|---|---|---|---|---|---|---|---|
| method\ dimension | 10 | 20 | 30 | 40 | 50 | 10 | 20 | 30 | 40 | 50 |
| | | | | Classification accuracy | | | | | | |
| rMMD-c | **0.610** | 0.596 | **0.556** | 0.526 | 0.511 | 0.668 | 0.660 | 0.647 | 0.619 | 0.588 |
| rMMD-k | 0.604 | **0.598** | 0.552 | **0.544** | **0.530** | **0.701** | **0.688** | **0.671** | **0.644** | **0.623** |
| n-Lasso | 0.505 | 0.498 | 0.503 | 0.500 | 0.502 | 0.509 | 0.504 | 0.502 | 0.502 | 0.503 |
| all | 0.568 | 0.516 | 0.500 | 0.500 | 0.504 | 0.639 | 0.544 | 0.516 | 0.503 | 0.502 |
| | | | | Test power | | | | | | |
| rMMD-t | 0.988 | **0.256** | **0.124** | **0.080** | 0.048 | 0.996 | **0.988** | **0.844** | **0.572** | **0.356** |
| MMD | **1.000** | 0.124 | 0.072 | 0.076 | **0.068** | **1.000** | 0.752 | 0.336 | 0.212 | 0.180 |

test set consists of 32 samples from each group. Feature filtering is conducted by selecting genes with variation greater than the 85%-quantile of the variations across all genes (Hahne et al., 2008), resulting in 1,895 genes for further analysis. Given that 1,895 dimensions are still too high for Lasso, we first reduce the dimensionality using the Sure Independence Screening (SIS) method (Fan & Song, 2009) before applying Lasso. The classification accuracy on the test set is presented in Table 4.

**Experimental Results of Table 2, Table 7 and Figure 2.** In Figure 2, $I^*$ represents the number of true important variables. In binary classification, our method demonstrated robust performance, achieving best values for both False Discovery Proportion (FDP) and accuracy across all cases, indicating its superiority over the Lasso method. Except for Case 1.f, the classification accuracy of our method remains stable as the dimension increases, showcasing its robustness and resilience in high-dimensional scenarios. For the two-sample tests, Table 7 indicates that the test power after variable selection consistently outperforms the original MMD test without variable selection in nearly all cases, except in instances where both methods show poor performance, with power close to the nominal size. Notably, in Case 2.e, the rMMD-t method demonstrates exceptional performance, effectively distinguishing between the two samples by selecting the relevant variables.

**Experimental Results of Table 3.** In binary classification, our method achieved the highest classification accuracy, while models built using variables selected by Lasso performed worse than those constructed with all variables. This indicates that our method effectively captures correlation differences, whereas Lasso fails completely in this scenario. Additionally, rMMD-t nearly outperforms MMD in two-sample tests, further emphasizing the necessity of variable selection for MMD tests.

Table 4: Classification accuracy in real data.

| Method | Accuracy |
|---|---|
| rMMD-c | **0.594** |
| n-Lasso | 0.484 |
| all | 0.500 |

**Experimental Results of Table 4.** From Table 4, rMMD-c achieves the highest accuracy, while n-Lasso's accuracy falls below 0.5, highlighting the effectiveness of our method and indicating that Lasso may no longer be suitable for variable selection in this dataset. Furthermore, this suggests that the variables selected by rMMD-c provide additional information that enhances the distinction between recurrence and non-recurrence tumor samples.

## 5 CONCLUSION

This paper introduces a variable selection method based on the MMD framework. Our approach assigns a weight to each variable, and the importance of variables is measured by solving an optimization problem to obtain optimal weights. We then propose an object-oriented algorithm to select variables for specific tasks, and the proposed accelerated algorithm significantly improves computational efficiency. Numerical experiments demonstrate the reliability of the proposed method. Although our approach is developed within the MMD framework, its underlying principles may extend to other frameworks, such as the Wasserstein distance. Future research will explore these possibilities.

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

# A APPENDIX

## A.1 ALGORITHM IN SECTION 3

### A.1.1 IMPORTANT VARIABLE SELECTION ALGORITHM IN BINARY CLASSIFICATION

---

**Algorithm 2** Regularized MMD for variable selection algorithm in binary classification

---

**Require:**

Samples $\mathfrak{X}_n, \mathfrak{Y}_m$, a sequence of tuning parameters $\lambda_1, \cdots, \lambda_q$;

#Phase 1: select the optimal tuning parameter.

**For** $j = 1, \ldots, q$ **do**

Select $\lambda_j$, obtained $\widehat{\mathbf{w}}_{\lambda_j}$ by solving equation 3.

Split the samples $\mathfrak{X}_{\widehat{\mathbf{w}}_{\lambda_j}}$ and $\mathfrak{Y}_{\widehat{\mathbf{w}}_{\lambda_j}}$ that contain only selected important variables by $\widehat{\mathbf{w}}_{\hat{\lambda}}$ into $K$ equal-sized subsets.

**For** $k = 1, \ldots, K$ **do**

Use the $k$-th subsets as the test set $\mathfrak{X}_{\widehat{\mathbf{w}}_{\lambda_j}}^{te,k}$ and $\mathfrak{Y}_{\widehat{\mathbf{w}}_{\lambda_j}}^{te,k}$.

Use the remaining $K - 1$ folds as the training set $\mathfrak{X}_{\widehat{\mathbf{w}}_{\lambda_j}}^{tr,k}$ and $\mathfrak{Y}_{\widehat{\mathbf{w}}_{\lambda_j}}^{tr,k}$.

Train a classifier $\hat{g}^{j,k}(x)$ based on training set.

Compute classification error $E_{\lambda_j}^k$ of the $\hat{g}^{j,k}(x)$ on the test set.

**End for**

Compute the average misclassification error $\ell(\lambda_j)$:

$$\ell(\lambda_j) = \frac{1}{K} \sum_{k=1}^{K} E_{\lambda_j}^k$$

**End for**

Let $\hat{\lambda} = \underset{\lambda_j, j=1,\cdots,q}{\operatorname{argmin}} \ell(\lambda_j)$, $\mathfrak{X}_{\widehat{\mathbf{w}}_{\hat{\lambda}}}$ and $\mathfrak{Y}_{\widehat{\mathbf{w}}_{\hat{\lambda}}}$ represents the samples that contain only selected important variables by $\widehat{\mathbf{w}}_{\hat{\lambda}}$.

#Phase 2: train a classifier $\hat{g}^*(x)$ based on $\mathfrak{X}_{\widehat{\mathbf{w}}_{\hat{\lambda}}}$ and $\mathfrak{Y}_{\widehat{\mathbf{w}}_{\hat{\lambda}}}$.

**Ensure:**

$\widehat{\mathbf{w}}_{\hat{\lambda}}, \hat{g}^*(x)$.

---

### A.1.2 DIFFERENCE OF CONVEX FUNCTIONS ALGORITHM AND MIRROR GRADIENT DECENT ALGORITHM

Now, we introduce the difference of convex functions algorithm. If $f(\cdot)$ is a convex function, the objective function equation 3 can be expressed as the difference between two convex functions:

$$-\widehat{\mathrm{MMD}}_{f,\mathbf{w}}^2 + \lambda \sum_{r=1}^{p} \omega_r^2 = g_1(\mathbf{w}) - g_2(\mathbf{w}),$$

where

$$g_1(\mathbf{w}) = \lambda \sum_{r=1}^{p} \omega_r^2 + 2n^{-1}m^{-1} \sum_{i=1}^{n} \sum_{j=1}^{m} f(\|\boldsymbol{X}_i - \boldsymbol{Y}_j\|_{\mathbf{w}}^2/\gamma)$$

$$g_2(\mathbf{w}) = \{n(n-1)\}^{-1} \sum_{1 \le i_1 \neq i_2 \le n} f(\|\boldsymbol{X}_{i_1} - \boldsymbol{X}_{i_2}\|_{\mathbf{w}}^2/\gamma)$$

$$+ \{m(m-1)\}^{-1} \sum_{1 \le j_1 \neq j_2 \le m} f(\|\boldsymbol{X}_{i_1} - \boldsymbol{X}_{i_2}\|_{\mathbf{w}}^2/\gamma).$$

According to the difference of convex functions algorithm (Thi & Dinh, 2018), we first perform a first-order Taylor expansion of $g_2(\mathbf{w})$ at $\mathbf{w} = \mathbf{w}^0$, where $\mathbf{w}^0$ is an initial value. This transforms the

objective function into:

$$\arg\min_{\mathbf{w}\in\boldsymbol{\Omega}}\left\{g_1(\mathbf{w}) - \left\{\left.\frac{\partial g_2(\mathbf{w})}{\partial\mathbf{w}}\right|_{\mathbf{w}=\mathbf{w}^0}\right\}^{\top}\mathbf{w}\right\}. \tag{7}$$

Note that equation 7 is a convex function, allowing us to employ a convex optimization algorithm for its solution. We choose the mirror descent algorithm, utilizing the Bregman distance defined as $B(\mathbf{x},\mathbf{y}) = \sum_{r=1}^{p} x_r \log(x_r/y_r)$ to address this convex optimization problem. Since the mirror descent algorithm is iterative, it provides only an approximate solution, which we use as the solution for equation 7. After completing one iteration, we denote the solution of equation 7 as $\mathbf{w}^1$. We then replace $\mathbf{w}^0$ in equation 7 with $\mathbf{w}^1$ and resolve the equation to obtain $\mathbf{w}^2$. This process is repeated until we reach the specified number of iterations. The detailed steps are outlined in Algorithm 3.

---

**Algorithm 3** Difference of convex functions algorithm

---

1: Input: initialize $\mathbf{w}^0 = \mathbf{1}$, the iteration step $T$ and $T_1$, the iteration step size $\alpha_{T_1}$.
2: **for** $t = 0$ to $T$ **do**
3:     Compute $\nabla\mathbf{g}_2^t = \left.\frac{\partial g_2(\mathbf{w})}{\partial\mathbf{w}}\right|_{\mathbf{w}=\mathbf{w}^t}$.
4:     $\mathbf{w}^{t+1} = \arg\min_{\mathbf{w}\in\boldsymbol{\Omega}}\left\{g_1(\mathbf{w}) - (\nabla\mathbf{g}_2^t)^{\top}\mathbf{w}\right\}$,
    where obtain $\mathbf{w}^{t+1}$ by Algorithm 4 to solve, the initial value is $\mathbf{w}^t$, the iteration step is $T_1$, and the iteration step size is $\alpha_{T_1}$, for Algorithm 4.
5: **end for**
6: Output: $\mathbf{w}^T$.

---

For a $p$-dimensional convex function $g(\mathbf{w})$, we introduce the mirror descent algorithm (Amir, 2017) in Algorithm 4 to solve the optimization problem $\arg\min_{\mathbf{w}\in\boldsymbol{\Omega}} g(\mathbf{w})$.

---

**Algorithm 4** Mirror gradient decent algorithm with Bregman distance $B(\mathbf{x},\mathbf{y}) = \sum_{r=1}^{p} x_r \log(x_r/y_r)$

---

1: Input: initialize $\mathbf{w}^0$, the iteration step $T$, and the iteration step size $\alpha_T$.
2: **for** $t = 0$ to $T$ **do**
3:     Compute $\nabla\mathbf{g}^t = \left.\frac{\partial g(\mathbf{w})}{\partial\mathbf{w}}\right|_{\mathbf{w}=\mathbf{w}^t}$.
4:     $\omega_r^{t+1} = p\frac{\omega_r^t - \exp\left\{-\alpha_T\nabla\mathbf{g}_r^t\right\}}{\sum_{k=1}^{p}\omega_k^t - \exp\{-\alpha_T\nabla\mathbf{g}_k^t\}}$, $r = 1, \cdots, p$, and $\mathbf{w}^{t+1} = (\omega_1^{t+1}, \cdots, \omega_p^{t+1})$.
5: **end for**
6: Output: $\mathbf{w}^T$.

---

## A.2    THE ABLATION STUDY ABOUT REGULARIZATION TERM

Now, we conduct an ablation study about regularization term. We will compare the impact of the presence or absence of a regularization term in the objective function on the selection of important variables in a binary classification problem.

The experiment setting is as follows: $\mathfrak{X} \sim N(\mu, \Sigma_1)$ and $\mathfrak{Y} \sim N(\mu, \Sigma_2)$, with $\mu = \mathbf{0}_p$, $\Sigma_2 = (0.2)^{|i-j|}1 \le i, j \le p$, and $\Sigma_2 = \Sigma_0 + \Sigma_1$ where $\Sigma_0 = \mathrm{diag}((10, 10, 1, 1), \mathbf{0}_{p-4})$. The train sample size is $n = m = 200$, and the test sample size is also $n = m = 100$. We compute accuracy on the test samples, the selection rates of $X_1$, $X_2$, $X_3$, and $X_4$. The results are presented in Table 5.

## A.3    ESITIMATOR OF MMD VARIANCE

The $\hat{\sigma}_{f,\widehat{\mathbf{w}}_\lambda}$ need to be estimated, according to Sutherland et al. (2017), when $n = m$,

$$\hat{\sigma}_{f,\widehat{\mathbf{w}}_\lambda} = \frac{4}{n^3}\sum_{i=1}^{n}\left(\sum_{j=1}^{n} h_{f,\widehat{\mathbf{w}}_\lambda}\left(\boldsymbol{X}_i, \boldsymbol{X}_j; \boldsymbol{Y}_i, \boldsymbol{Y}_j\right)\right)^2 - \frac{4}{n^4}\left(\sum_{i=1}^{n}\sum_{j=1}^{n} h_{f,\widehat{\mathbf{w}}_\lambda}\left(\boldsymbol{X}_i, \boldsymbol{X}_j; \boldsymbol{Y}_i, \boldsymbol{Y}_j\right)\right)^2.$$

where $h_{f,\widehat{\mathbf{w}}_\lambda}(\boldsymbol{x}, \boldsymbol{x}'; \boldsymbol{y}, \boldsymbol{y}') = f(\|\boldsymbol{x} - \boldsymbol{x}'\|^2_{\widehat{\mathbf{w}}_\lambda}/\gamma) + f(\|\boldsymbol{y} - \boldsymbol{y}'\|^2_{\widehat{\mathbf{w}}_\lambda}/\gamma) - f(\|\boldsymbol{x} - \boldsymbol{y}'\|^2_{\widehat{\mathbf{w}}_\lambda}/\gamma) - f(\|\boldsymbol{x}' - \boldsymbol{y}\|^2_{\widehat{\mathbf{w}}_\lambda}/\gamma)$. When $n \ne m$, please refer to Sutherland & Deka (2022).

Table 5: The selection rate of important variables.

| method | $X_1$ | $X_2$ | $X_3$ | $X_4$ |
|---|---|---|---|---|
| | $p = 50$ | | | |
| Non-regularization | 1.000 | 1.000 | 0.070 | 0.065 |
| Regularization | 1.000 | 1.000 | 0.130 | 0.170 |
| | $p = 100$ | | | |
| Non-regularization | 0.995 | 1.000 | 0.135 | 0.115 |
| Regularization | 0.995 | 1.000 | 0.225 | 0.255 |

## A.4 COMPARISON BETWEEN THE ACCELERATED METHOD AND THE ORIGINAL METHOD

Now, we conduct a simulation experiment to compare the results of the accelerated method with the original method in binary classification problems.

The experiment setting is as follows: $\mathfrak{X} \sim N(\mu, \Sigma_1)$ and $\mathfrak{Y} \sim N(\mu, \Sigma_2)$, with $\mu = \mathbf{0}_p$, $\Sigma_2 = (0.2)^{|i-j|} 1 \le i, j \le p$, and $\Sigma_2 = \Sigma_0 + \Sigma_1$ where $\Sigma_0 = \text{diag}(\mathbf{3}_4, \mathbf{0}_{p-4})$. The train sample size is $n = m = 100$, and the test sample size is also $n = m = 100$. We compute accuracy on the test samples, the selection rates of $X_1$, $X_2$, $X_3$, and $X_4$, and FDP.

Table 6: Accuracy, selection rate of important variables and FDP.

| method | Accuracy | $X_1$ | $X_2$ | $X_3$ | $X_4$ | FDP |
|---|---|---|---|---|---|---|
| | | $p = 30$ | | | | |
| Original | 0.726 | 0.940 | 0.980 | 0.940 | 0.980 | 0.014 |
| Accelerated | 0.714 | 0.760 | 0.830 | 0.860 | 0.800 | 0.044 |
| | | $p = 60$ | | | | |
| Original | 0.721 | 0.990 | 1.000 | 0.980 | 0.980 | 0.033 |
| Accelerated | 0.695 | 0.780 | 0.780 | 0.820 | 0.830 | 0.113 |

From Table 6, we can see that the accelerated method has a slightly lower important variable selection rate compared to the original method; however, it has almost no impact on classification accuracy. This suggests that while the accelerated method may miss some variables, it still captures most of the important ones and achieves similar classification accuracy. Considering the significant computational time advantage of the accelerated method, we recommend prioritizing its use.

## A.5 DETAILS OF THE NUMERICAL EXPERIMENTS

First, we provide the following symbols:

**Symbols and their descriptions**

| Symbol | Description |
|---|---|
| $\mathbf{I}^*$ | The number of important variables. |
| $\text{seq}(a, b, t)$ | The arithmetic sequence vector from $a$ to $b$ with a common difference of $t$, for example $\text{seq}(1, 4, 1) = (1, 2, 3, 4)$. |
| $\boldsymbol{X} = (X_1, \cdots, X_p)^\top$ | Representation of the population of samples $\mathfrak{X}_n$ |
| $\boldsymbol{Y} = (Y_1, \cdots, Y_p)^\top$ | Representation of the population of samples $\mathfrak{Y}_m$ |

### A.5.1 SETTING IN SYNTHETIC DATA 1

In Synthetic Data 1, the number of important variables is 4 or 8. In binary classification, the training and test sample sizes are set to $n_{tr} = m_{tr} = 200$ and $n_{te} = m_{te} = 100$, respectively. For two-sample tests, the sample size is $n = m = 200$. These experiments were repeated 250 times.

For binary classification, the samples $\mathfrak{X}_n$ and $\mathfrak{Y}_m$ is obatained by the following six cases.

Case 1.a. $\boldsymbol{X} \sim N(\mu_1, \Sigma)$, $\boldsymbol{Y} \sim N(\mu_2, \Sigma)$, $\mu_1 = \boldsymbol{0}_p$, , $\Sigma = \{(0.2)^{|i-j|}\}_{1 \leq i,j \leq p}$,

$$\mu_2 = \{\text{seq}(1/\mathbf{I}^*, 1, 1/\mathbf{I}^*), \boldsymbol{0}_{p-\mathbf{I}^*}\}$$

Case 1.b. $\boldsymbol{X} \sim N(\boldsymbol{0}_p, \Sigma_1)$, $\boldsymbol{Y} \sim N(\boldsymbol{0}_p, \Sigma_2)$, $\Sigma_2 = \{(0.2)^{|i-j|}\}_{1 \leq i,j \leq p}$, $\Sigma_2 = \Sigma_0 + \Sigma_1$,

$$\Sigma_0 = \text{diag}(\text{seq}(1/\mathbf{I}^*, 1, 1/\mathbf{I}^*), \boldsymbol{0}_{p-\mathbf{I}^*})$$

Case 1.c. $X_r \sim \chi^2(1)$, $Y_r \sim \chi^2(V_r)$, $X_r$ and $Y_r$ both are mutually independent,

$$(V_1, \cdots, V_p) = \{\text{seq}(1 + 4/\mathbf{I}^*, 5, 4/\mathbf{I}^*), \boldsymbol{1}_{p-\mathbf{I}^*}\}.$$

Case 1.d. $\boldsymbol{X} \sim N(\boldsymbol{0}_p, \Sigma)$, $\Sigma = \text{diag}(\boldsymbol{3}_p)$, $Y_r$ are mutually independent,

$$Y_r \sim \exp(1/3), \ r \leq \mathbf{I}^*; \ Y_r \sim N(1, 3), r > \mathbf{I}^*.$$

Case 1.e. $\boldsymbol{X} \sim N(\boldsymbol{0}_p, \Sigma_1)$, $\Sigma_1 = \text{diag}(\boldsymbol{1}_p)$, $Y_r$ are mutually independent. In addition, let $z \sim B(1, 0.5)$, $z$ and $Y_r$ are independent.

$$Y_r \sim zN(-2, 1) + (1 - z)N(2, 1), \ r \leq \mathbf{I}^*; \ Y_r \sim N(1, 1), \ r > \mathbf{I}^*.$$

Case 1.f. $X_r \sim U(0, 1)$, $X_r$ and $Y_r$ both are mutually independent,

$$Y_r \sim \text{Beta}(15, 15), \ r \leq \mathbf{I}^*; \ Y_r \sim \text{U}(0, 1), \ r > \mathbf{I}^*.$$

For two-sample tests, the samples $\mathfrak{X}_n$ and $\mathfrak{Y}_m$ is illustrated by the following six cases.

The settings of Case 2.a, Case 2.d, Case 2.e, Case 2.f are the same as in Case 1.b, Case 1.d, Case 1.e, and Case 1.f.

Case 2.b. $X_r \sim Cauchy(0, 1)$, $Y_r \sim Cauchy(V_r, 1)$, $X_r$ and $Y_r$ both are mutually independent,

$$(V_1, \cdots, V_p) = \{\text{seq}(4/\mathbf{I}^*, 4, 4/\mathbf{I}^*), \boldsymbol{0}_{p-\mathbf{I}^*}\}.$$

Case 2.c. $X_r \sim Cauchy(0, 1)$, $Y_r \sim Cauchy(0, V_r)$, $X_r$ and $Y_r$ both are mutually independent,

$$(V_1, \cdots, V_p) = \{\text{seq}(1, 9, 8/\mathbf{I}^*), \boldsymbol{0}_{p-\mathbf{I}^*}\}.$$

### A.5.2 SETTING IN SYNTHETIC DATA 2

In Synthetic Data 2, involve either 3 or 6 important variables. For binary classification, the training and test sample sizes are set to $n_{tr} = m_{tr} = 300$ and $n_{te} = m_{te} = 100$, respectively. For two-sample tests, the sample size is $n = m = 300$. Each experiment was repeated 250 times. The samples $\mathfrak{X}_n$ and $\mathfrak{Y}_m$ is obtained by the following distributions.

$\boldsymbol{X} \sim N(\mu, \Sigma_1)$, $\boldsymbol{Y} \sim N(\mu, \Sigma_2)$, where $\mu = (\boldsymbol{0}_p)$, $\Sigma_1 = \text{diag}(\boldsymbol{1}_p)$,

$$\Sigma_2 = \begin{pmatrix} \Sigma_{1,1} & \boldsymbol{0} \\ \boldsymbol{0} & \Sigma_{2,2} \end{pmatrix}, \text{with } \Sigma_{1,1} = \begin{pmatrix} 1 & \cdots & 0.6 \\ \vdots & \ddots & \vdots \\ 0.6 & \cdots & 1 \end{pmatrix}_{\mathbf{I}^* \times \mathbf{I}^*}, \Sigma_{2,2} = \text{diag}(\boldsymbol{1}_{p-\mathbf{I}^*}).$$

### A.5.3 TYPE-I ERROR STUDY

We now conduct a simulation experiment to demonstrate the performance of our method under the null hypothesis holding. The experiment setting is as follows: $\mathfrak{X} \sim N(\mu, \Sigma)$ and $\mathfrak{Y} \sim N(\mu, \Sigma)$, with $\mu = \boldsymbol{0}_p$, $\Sigma = (0.2)^{|i-j|} 1 \leq i, j \leq p$, The sample size is $n = m = 200$. The significance level $\alpha = 0.05$ We compute empirical Type-I error.

From the Table 9, we can observe that under the null hypothesis holding, our method fluctuates around the specified significance level $\alpha = 0.05$, indicating that the I-Type error of our method is effectively controlled.

Table 7: Test power in Synthetic Data 1.

| $I^*$ | | 4 | | | | | 8 | | | |
| method\ dimension | 50 | 100 | 150 | 200 | 250 | 50 | 100 | 150 | 200 | 250 |
| | | | | | Test power | | | | | |
| | | | | | Case 2.a | | | | | |
| rMMD-t | **0.998** | **0.968** | **0.696** | **0.514** | **0.416** | **1.000** | **1.000** | **1.000** | **0.966** | **0.840** |
| MMD | 0.856 | 0.282 | 0.122 | 0.084 | 0.090 | 1.000 | 0.880 | 0.486 | 0.268 | 0.202 |
| | | | | | Case 2.b | | | | | |
| rMMD-t | **0.967** | **0.521** | **0.323** | **0.291** | **0.192** | **0.985** | **0.613** | **0.386** | **0.316** | **0.225** |
| MMD | 0.110 | 0.070 | 0.068 | 0.072 | 0.058 | 0.160 | 0.060 | 0.060 | 0.050 | 0.030 |
| | | | | | Case 2.c | | | | | |
| rMMD-t | **0.988** | **0.872** | **0.684** | **0.564** | **0.388** | **1.000** | **0.956** | **0.880** | **0.764** | **0.572** |
| MMD | 0.776 | 0.276 | 0.156 | 0.128 | 0.084 | 0.984 | 0.612 | 0.368 | 0.228 | 0.148 |
| | | | | | Case 2.d | | | | | |
| rMMD-t | **0.976** | **0.592** | **0.224** | **0.120** | **0.124** | **1.000** | **0.980** | **0.852** | **0.584** | **0.312** |
| MMD | 0.516 | 0.144 | 0.112 | 0.112 | 0.084 | 0.996 | 0.672 | 0.348 | 0.200 | 0.168 |
| | | | | | Case 2.e | | | | | |
| rMMD-t | **1.000** | **1.000** | **1.000** | **0.996** | **0.944** | **1.000** | **1.000** | **1.000** | **1.000** | **1.000** |
| MMD | 1.000 | 0.760 | 0.412 | 0.260 | 0.200 | 1.000 | 1.000 | 1.000 | 0.904 | 0.704 |
| | | | | | Case 2.f | | | | | |
| rMMD-t | **0.636** | **0.100** | **0.068** | 0.036 | **0.052** | **1.000** | **0.648** | **0.192** | 0.064 | 0.044 |
| MMD | 0.168 | 0.060 | 0.044 | **0.044** | 0.044 | 0.596 | 0.120 | 0.068 | **0.076** | **0.052** |

Table 8: The standard deviations of classification accuracy and test power in Synthetic Data 2.

| $I^*$ | | 3 | | | | | 6 | | | |
| method\ dimension | 10 | 20 | 30 | 40 | 50 | 10 | 20 | 30 | 40 | 50 |
| | | | | Classification accuracy | | | | | | |
| rMMD-c | 0.059 | 0.054 | 0.047 | 0.050 | 0.042 | 0.035 | 0.041 | 0.038 | 0.047 | 0.059 |
| rMMD-k | 0.061 | 0.047 | 0.055 | 0.038 | 0.040 | 0.029 | 0.034 | 0.037 | 0.039 | 0.038 |
| n-Lasso | 0.032 | 0.031 | 0.033 | 0.034 | 0.031 | 0.033 | 0.036 | 0.034 | 0.033 | 0.032 |
| all | 0.049 | 0.043 | 0.045 | 0.036 | 0.032 | 0.040 | 0.043 | 0.038 | 0.041 | 0.037 |

### A.5.4 FDP FOR CLASSIFICATION

Figure 3 show the FDP varies with the dimensionality in Case 1.e and Case 1.f.

### A.6 PROOFS OF THE THEOREMS

**Lemma 1.** *Let $\{X_1, \ldots, X_n\}$ and $\{Y_1, \ldots, Y_m\}$ be independent and identically distributed $p$-dimensional random variables from distributions $\mathbf{F}$ and $\mathbf{G}$, respectively. Let $\varphi$ be a symmetric kernel function, and define*

$$U_{n,m} = \frac{1}{nm} \sum_{i=1}^{n} \sum_{j=1}^{m} \varphi\left(X_i, Y_j\right),$$

*Denote $\vartheta = \mathbb{E}U_{n,m} = \mathbb{E}\left\{\varphi\left(X_1, Y_1\right)\right\}$, and $n_0 = \min\{n, m\}$.*

*(i) If the kernel function $\varphi$ satisfies $C_{\varphi 1} \leq \varphi \leq C_{\varphi 2}$ for some constants $C_{\varphi 1} \leq C_{\varphi 2}$, then for any $\epsilon > 0$, we have*

$$\mathrm{P}\left(|U_{n,m} - \vartheta| \geq \epsilon\right) \leq 2 \exp\left\{-2n_0\epsilon^2 / \left(C_{\varphi 2} - C_{\varphi 1}\right)^2\right\}.$$

*(ii) If there exists a positive constant $s > 4$ such that $C_1 = \mathbb{E}\{|\varphi(X_1, Y_1)|^s\} < \infty$, then for any $\epsilon > 0$,*

$$\mathrm{P}(|U_{n,m} - \vartheta| \geq \epsilon) \leq 2 \exp(-2^{-1}n_0\epsilon^2/\zeta_1^2) + nmC_1/\zeta_1^s.$$

Table 9: Type-I error

| method/dimension | 50 | 100 | 150 | 200 | 250 |
|---|---|---|---|---|---|
| rMMD-t | 0.056 | 0.049 | 0.044 | 0.040 | 0.052 |
| MMD | 0.040 | 0.052 | 0.048 | 0.044 | 0.028 |

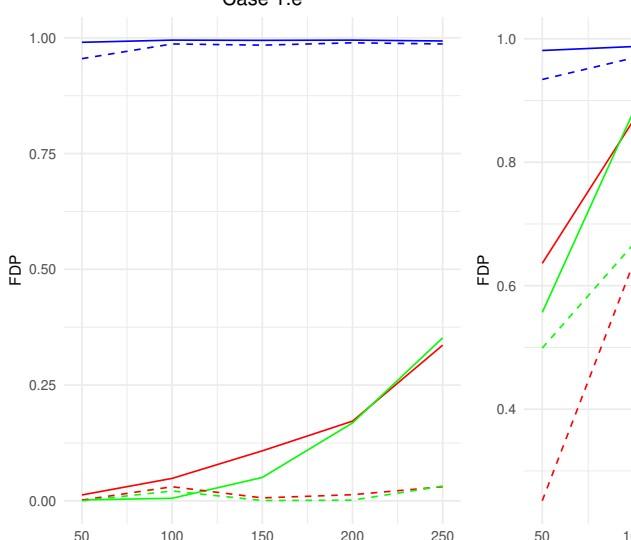

Figure 3: The averages FDP (y-axis) over 250 replications across different dimensions (x-axis) in binary classification. The solid curves correspond to cases with 4 important variables, and the dashed curves represent cases with 8 important variables. The red curves represent our method, and the green curves represent rMMD-c, while the blue lines depict the n-Lasso method.

*where $\zeta_1$ satisfies $\mathbb{E}[\{\varphi(\boldsymbol{X}_1, \boldsymbol{Y}_1)\}^2]\mathbb{E}\{|\varphi(\boldsymbol{X}_1, \boldsymbol{Y}_1)|^s\}/\zeta_1^s < \epsilon^2/16$.*

*Proof.* (i) Denote $\psi(\boldsymbol{x}_1, \ldots, \boldsymbol{x}_n; \boldsymbol{y}_1, \ldots, \boldsymbol{y}_m) = n_0^{-1}\sum_{i=1}^{n_0}\varphi(\boldsymbol{x}_i, \boldsymbol{y}_i)$, the two-sample $U$-statistic $U_{n,m}$ can be represented as

$$U_{n,m} = \frac{1}{n!m!}\sum_* \psi(\boldsymbol{X}_{i_1}, \ldots, \boldsymbol{X}_{i_n}; \boldsymbol{Y}_{j_1}, \ldots, \boldsymbol{Y}_{j_m}),$$

where $n!$ and $m!$ are factorial, and the summation $\sum_*$ is carried out over all permutation $(i_1, \ldots, i_n)$ and $(j_1, \ldots, j_m)$ of the numbers $(1, \ldots, n)$ and $(1, \ldots, m)$, respectively. As the exponential function is convex, by Jensen's inequality, we have for any $\xi > 0$,

$$\mathbb{E}\{\exp(\xi U_{n,m})\} \leq (n!)^{-1}(m!)^{-1}\sum_* \mathbb{E}[\exp\{\xi\psi(\boldsymbol{X}_{i_1}, \ldots, \boldsymbol{X}_{i_n}; \boldsymbol{Y}_{j_1}, \ldots, \boldsymbol{Y}_{j_m})\}]$$

$$= \mathbb{E}[\exp\{\xi\psi(\boldsymbol{X}_{1_1}, \ldots, \boldsymbol{X}_{1_n}; \boldsymbol{Y}_{1_1}, \ldots, \boldsymbol{Y}_{1_m})\}]$$

$$= (\mathbb{E}[\exp\{n_0^{-1}\xi\varphi(\boldsymbol{X}_1, \boldsymbol{Y}_1)\}])^{n_0},$$

The last equation holds because $\psi(\boldsymbol{X}_{1_1}, \ldots, \boldsymbol{X}_{1_n}; \boldsymbol{Y}_{1_1}, \ldots, \boldsymbol{Y}_{1_m}) = n_0^{-1}\sum_{i=1}^{n_0}\varphi(\boldsymbol{X}_i; \boldsymbol{Y}_i)$ is the summation of $n_0$ i.i.d random variables. According to Markov's inequality and Lemma 8.1.1 of Koroljuk & Borovskich (1994),

$$\mathrm{P}(U_{n,m} - \vartheta \geq \epsilon) \leq \mathbb{E}[\exp\{\xi(U_{n,m} - \vartheta)\}]\exp(-\xi\epsilon)$$

$$\leq (\mathbb{E}[\exp\{n_0^{-1}\xi\varphi(\boldsymbol{X}_1; \boldsymbol{Y}_1)\}])^{n_0}\exp(-\xi\vartheta)\exp(-\xi\epsilon)$$

$$= (\mathbb{E}[\exp\{n_0^{-1}\xi(\varphi(\boldsymbol{X}_1; \boldsymbol{Y}_1) - \vartheta)\}])^{n_0}\exp(-\xi\epsilon)$$

$$\leq \exp\{-\xi\epsilon + \xi^2(C_{\varphi 2} - C_{\varphi 1})^2/(8n_0)\}.$$

Choosing $\xi = 4n_0\epsilon/(C_{\varphi 2} - C_{\varphi 1})^2$, we arrive at

$$\mathrm{P}(U_{n,m} - \vartheta \geq \epsilon) \leq \exp\{-2n_0\epsilon^2/(C_{\varphi 2} - C_{\varphi 1})^2\}.$$

Then by the symmetric property of two-sample $U$-statistics, we complete the proof of (i).

(ii) Decompose the kernel function $\varphi$ as $\varphi = \varphi I(\varphi \leq \zeta_1) + \varphi I(\varphi > \zeta_1)$, where $\zeta_1 > 0$ will be specified later. Then $U_{n,m}$ can be written as

$$U_{n,m} = n^{-1}m^{-1} \sum_{i=1}^{n} \sum_{j=1}^{m} \varphi(\boldsymbol{X}_i; \boldsymbol{Y}_j) I(\varphi(\boldsymbol{X}_i; \boldsymbol{Y}_j) \leq \zeta_1)$$

$$+ n^{-1}m^{-1} \sum_{i=1}^{n} \sum_{j=1}^{m} \varphi(\boldsymbol{X}_i; \boldsymbol{Y}_j) I(\varphi(\boldsymbol{X}_i; \boldsymbol{Y}_j) > \zeta_1)$$

$$:= U_{n,m,1} + U_{n,m,2},$$

and $\vartheta$ can be decomposed as $\vartheta = \mathbb{E}U_{n,m} = \mathbb{E}U_{n,m,1} + \mathbb{E}U_{n,m,2} := \vartheta_1 + \vartheta_2$.

First, according to the result of (i), it is easy to obtain that for any $\epsilon > 0$,

$$\mathrm{P}(|U_{n,m,1} - \vartheta_1| \geq \epsilon/2) \leq 2\exp\{-2^{-1}n_0\epsilon^2/\zeta_1^2\}.$$

Furthermore, by utilizing Cauchy-Schwartz and Markov's inequality,

$$\vartheta_2^2 \leq \mathbb{E}\{\varphi(\boldsymbol{X}_1; \boldsymbol{Y}_1)\}^2 \mathrm{P}(\varphi(\boldsymbol{X}_1; \boldsymbol{Y}_1) > \zeta_1) \leq \mathbb{E}\{\varphi(\boldsymbol{X}_1; \boldsymbol{Y}_1)\}^2 \mathbb{E}[|\varphi(\boldsymbol{X}_1; \boldsymbol{Y}_1)|^s]/\zeta_1^s.$$

As $\mathbb{E}[|\varphi(\boldsymbol{X}_1; \boldsymbol{Y}_1)|^s] < \infty$, for any $\epsilon > 0$, $\zeta$ can be choose such that

$$\mathbb{E}[\{\varphi(\boldsymbol{X}_1, \boldsymbol{Y}_1)\}^2]\mathbb{E}\{|\varphi(\boldsymbol{X}_1, \boldsymbol{Y}_1)|^s\}/\zeta_1^s < \epsilon^2/16.$$

In this situation, $\vartheta_2 < \epsilon/4$, from which we get

$$\mathrm{P}(|U_{n,m,2} - \vartheta_2| \geq \epsilon/2) \leq \mathrm{P}(|U_{n,m,2}| \geq \epsilon/4).$$

If $|U_{n,m,2}| \geq \epsilon/4$ is true, there must exist some $i \in \{1, \ldots, n\}$ or $j \in \{1, \ldots, m\}$, such that $\varphi(\boldsymbol{X}_i; \boldsymbol{Y}_j) > \zeta_1$. This can be proved by contradiction. Suppose $\varphi(\boldsymbol{X}_i; \boldsymbol{Y}_j) \leq \zeta_1$ for all $1 \leq i \leq n$ and $1 \leq j \leq m$, then $U_{n,m,2} = 0$ which is contradicted with $|U_{n,m,2}| \geq \epsilon/4$.

According to Markov's inequality,

$$\mathrm{P}(\varphi(\boldsymbol{X}_1, \boldsymbol{Y}_1) > \zeta_1) \leq \mathbb{E}[|\varphi(\boldsymbol{X}_1; \boldsymbol{Y}_1)|^s]/\zeta_1^s \leq C_1/\zeta_1^s.$$

This leads to that

$$\mathrm{P}(|U_{n,m,2} - \vartheta_2| \geq \epsilon/2) \leq nm\,\mathrm{P}(\varphi(\boldsymbol{X}_1, \boldsymbol{Y}_1) > \zeta_1) \leq C_1 nm/\zeta_1^s.$$

Finally,

$$\mathrm{P}(|U_{n,m} - \vartheta| \geq \epsilon) \leq \mathrm{P}(|U_{n,m,1} - \vartheta_1| \geq \epsilon/2) + \mathrm{P}(|U_{n,m,2} - \vartheta_2| \geq \epsilon/2)$$

$$\leq 2\exp\{-2^{-1}n_0\epsilon^2/\zeta_1^2\} + C_1 nm/\zeta_1^s.$$

from which we complete the proof of this lemma.

$\square$

**Lemma 2.** *Let $\{\boldsymbol{X}_1, \ldots, \boldsymbol{X}_n\}$ be independent and identically distributed p-dimensional random variables from the distribution $\mathbf{F}$.*

$$U_n = \frac{2}{n(n-1)} \sum_{1 \leq i < j \leq n} h(\boldsymbol{X}_i, \boldsymbol{X}_j)$$

*is a U-statistic with a symmetric kernel function $h$. Let $\theta = \mathbb{E}U_n = \mathbb{E}\{h(\boldsymbol{X}_1, \boldsymbol{X}_2)\}$.*

*(i) If the kernel function $h$ satisfies $C_{h1} \leq h \leq C_{h2}$ for some constants $C_{h1} \leq C_{h2}$, then for any $\epsilon > 0$ and $n \geq 2$,*

$$\mathrm{P}(|U_n - \theta| \geq \epsilon) \leq 2\exp\left\{-2\lceil n/2 \rceil \epsilon^2/(C_{h2} - C_{h1})^2\right\}$$

*(ii) If there exists a positive constant $s > 4$ such that $C_2 = \mathbb{E}\{|h(\boldsymbol{X}_1, \boldsymbol{X}_2)|^s\} < \infty$, then for any $\epsilon > 0$ and $n \geq 2$,*

$$P(|U_n - \theta| \geq \epsilon) \leq 2\exp(-2^{-1}\lceil n/2\rceil\epsilon^2/\zeta_2^2) + n(n-1)C_2/\zeta_2^s.$$

*where $\zeta_2$ satisfies $\mathbb{E}[\{h(\boldsymbol{X}_1, \boldsymbol{X}_2)\}^2]\mathbb{E}\{|h(\boldsymbol{X}_1, \boldsymbol{X}_2)|^s\}/\zeta_2^s < \epsilon^2/16$.*

*Proof.* The proof of Lemma 2 is similar to the proof of Lemma 1. $\qquad\square$

*Proof of Theorem 1.* Firstly, let's examine the boundary of $P(|[\widehat{\mathrm{dMMD}}_f^2]^\top\mathbf{w} - [\mathrm{dMMD}_f^2]^\top\mathbf{w}| > \varepsilon)$, for any $\varepsilon > 0$. For $r = 1, \ldots, d$,

$$\mathrm{df}_r(\boldsymbol{X}_1, \boldsymbol{Y}_1) = \gamma^{-1}(X_r - Y_r)^2 f^{(1)}(\|\boldsymbol{X} - \boldsymbol{Y}\|^2/\gamma),$$

with $\boldsymbol{X}_1 = (X_{1,1}\ldots, X_{1,d})^\top$ and $\boldsymbol{Y}_1 = (Y_{1,1}, \ldots, Y_{1,d})^\top$. Under Assumption 2 and 3, for any $\gamma > 0$, the kernel function $\mathrm{df}_r(\boldsymbol{X}_1, \boldsymbol{Y}_1)$ satisfies

$$\delta_1 = \mathbb{E}\{|\mathrm{df}_r(\boldsymbol{X}_1, \boldsymbol{Y}_1)|^s\} \leq \gamma^{-s}B_1^s\mathbb{E}\left\{(X_{1,r} - Y_{1,r})^{2s}\right\} \leq 2^{2s}\gamma^{-s}B_1^s\mathbb{E}\left(|X_{1,r}|^{2s} + |Y_{1,r}|^{2s}\right) < \infty.$$

Similarly, $\delta_2 = \mathbb{E}\{|\mathrm{df}_r(\boldsymbol{X}_1, \boldsymbol{X}_2)|^s\} < \infty$ and $\delta_3 = \mathbb{E}\{|\mathrm{df}_r(\boldsymbol{Y}_1, \boldsymbol{Y}_2)|^s\} < \infty$. By Bonferroni's inequality, for any $\mathbf{w} \in \boldsymbol{\Omega}$ and $\varepsilon > 0$, we have

$$P\left(\left|\left[\widehat{\mathrm{dMMD}}_f^2\right]^\top\mathbf{w} - [\mathrm{dMMD}_f^2]^\top\mathbf{w}\right| \geq \varepsilon\right) = P\left(\left|\sum_{r=1}^{p}\left(\widehat{\mathrm{dMMD}}_f^{2(r)} - \mathrm{dMMD}_f^{2(r)}\right)\omega_r\right| \geq \varepsilon\right)$$

$$\leq \sum_{r=1}^{p}P\left(\left|\frac{1}{nm}\sum_{i=1}^{n}\sum_{j=1}^{m}\mathrm{df}_r(\boldsymbol{X}_i, \boldsymbol{Y}_j) - \mathbb{E}[\mathrm{df}_r(\boldsymbol{X}_1, \boldsymbol{X}_2)]\right| \geq \frac{\varepsilon}{4p\omega_r}\right)$$

$$+ \sum_{r=1}^{p}P\left(\left|\frac{1}{n(n-1)}\sum_{1\leq i_1\neq i_2\leq n}\mathrm{df}_r(\boldsymbol{X}_{i_1}, \boldsymbol{X}_{i_2}) - \mathbb{E}[\mathrm{df}_r(\boldsymbol{X}_1, \boldsymbol{X}_2)]\right| \geq \frac{\varepsilon}{4p\omega_r}\right)$$

$$+ \sum_{r=1}^{p}P\left(\left|\frac{1}{m(m-1)}\sum_{1\leq j_1\neq j_2\leq m}\mathrm{df}_r(\boldsymbol{Y}_{j_1}, \boldsymbol{Y}_{j_2}) - \mathbb{E}[\mathrm{df}_r(\boldsymbol{Y}_1, \boldsymbol{Y}_2)]\right| \geq \frac{\varepsilon}{4p\omega_r}\right).$$

According to Lemma 1 (ii) and Lemma 2 (ii), we have

$$P\left(\left|\left[\widehat{\mathrm{dMMD}}_f^2\right]^\top\mathbf{w} - [\mathrm{dMMD}_f^2]^\top\mathbf{w}\right| \geq \varepsilon\right)$$

$$\leq \sum_{r=1}^{p}\left\{2\exp(-32^{-1}p^{-2}\omega_r^{-2}n_0\varepsilon^2/\zeta^2) + nm\delta_1/\zeta^s\right.$$

$$+ 2\exp(-32^{-1}p^{-2}\omega_r^{-2}\lceil n/2\rceil\varepsilon^2/\zeta^2) + n(n-1)\delta_2/\zeta^s$$

$$\left. + 2\exp(-32^{-1}p^{-2}\omega_r^{-2}\lceil m/2\rceil\varepsilon^2/\zeta^2) + m(m-1)\delta_3/\zeta^s\right\}.$$

where $n_0 = \min\{n, m\}$ and $\zeta$ satisfies

$$\mathbb{E}[\{\mathrm{df}_r(\boldsymbol{X}_1, \boldsymbol{Y}_1)\}^2]\mathbb{E}\{|\mathrm{df}_r(\boldsymbol{X}_1, \boldsymbol{X}_2)|^s\}/\zeta^s < \varepsilon^2/(256p^2\omega_r^2),$$
$$\mathbb{E}[\{\mathrm{df}_r(\boldsymbol{X}_1, \boldsymbol{X}_2)\}^2]\mathbb{E}\{|\mathrm{df}_r(\boldsymbol{X}_1, \boldsymbol{X}_2)|^s\}/\zeta^s < \varepsilon^2/(256p^2\omega_r^2),$$
$$\mathbb{E}[\{\mathrm{df}_r(\boldsymbol{Y}_1, \boldsymbol{X}_2)\}^2]\mathbb{E}\{|\mathrm{df}_r(\boldsymbol{X}_1, \boldsymbol{Y}_2)|^s\}/\zeta^s < \varepsilon^2/(256p^2\omega_r^2).$$

For any fixed $c > 0$ and $0 < \kappa < 1/2 - 2/s$, let $\varepsilon = cN^{-\kappa}$ and choose $\zeta = N^\iota$ for some positive $\iota$ satisfying $\iota + \kappa < 1/2$ and $s\iota > 2$. It follows that for sufficiently large $N$, there exists a positive constant $c_1$ such that

$$\left(\left|\left[\widehat{\mathrm{dMMD}}_f^2\right]^\top\mathbf{w} - [\mathrm{dMMD}_f^2]^\top\mathbf{w}\right| \geq cN^{-\kappa}\right) \leq O\left(p\exp\left\{-c_1p^{-2}N^{1-2(\kappa+\iota)}\right\}\right) + O\left(pN^{2-s\iota}\right).$$

Let $Z_N(\mathbf{w}) = -\left[\widehat{\mathrm{dMMD}}_f^2\right]^\top \mathbf{w} + \lambda \sum_{r=1}^p \omega_r^2$ and $Z(\mathbf{w}) = -\left[\widehat{\mathrm{dMMD}}_f^2\right]^\top \mathbf{w} + \lambda \sum_{r=1}^p \omega_r^2$. Therefore, when $N \to \infty$,

$$P\left(\sup_{\mathbf{w} \in \boldsymbol{\Omega}} |Z_N(\mathbf{w}) - Z(\mathbf{w})| > \varepsilon\right) \leq \sup_{\mathbf{w} \in \boldsymbol{\Omega}} P\left(|Z_N(\mathbf{w}) - Z(\mathbf{w})| > \varepsilon\right) \longrightarrow 0.$$

It is easy to know that $\boldsymbol{\Omega}$ is a convex set, and $Z_N(\mathbf{w}), Z(\mathbf{w})$ are strongly convex functions, so the minimum point is unique. According to $\sup_{\mathbf{w} \in \boldsymbol{\Omega}} |Z_N(\mathbf{w}) - Z(\mathbf{w})| \xrightarrow{P} 0$ and the uniqueness of the minimum point, we have $\hat{\mathbf{w}}_\lambda \xrightarrow{P} \mathbf{w}_\lambda^*$. $\qquad\square$

*Proof of Theorem 2.* First, we show the Bregman distance used in the mirror descent algorithm,

$$B(\mathbf{x}, \mathbf{y}) = \sum_{r=1}^p x_r \log (x_r/y_r)$$

According to the iteration update formula of the mirror descent algorithm in Chapter 9 Amir (2017), given iteration step $T$ and an iteration step size $\alpha_T$, we have

$$\mathbf{w}^{t+1} = \underset{\mathbf{w} \in \boldsymbol{\Omega}}{\operatorname{argmin}} \left\{\sum_{r=1}^p \left(\alpha_T \widehat{\mathrm{dMMD}}_f^{2(r)} - 1 - \log\left(\omega_r^t\right)\right) \omega_r + \sum_{r=1}^p \omega_r \log \omega_r\right\}, \tag{8}$$

By similar proof in Amir (2017) Example 3.71, the optimal solution of (8), we have

$$\omega_r^{t+1} = p\left[\frac{\omega_r^t - \exp\left\{-\alpha_T(-\widehat{\mathrm{dMMD}}_f^{2(r)} + 2\lambda\omega_r^k)\right\}}{\sum_{k=1}^p \omega_k^t - \exp\left\{-\alpha_T(-\widehat{\mathrm{dMMD}}_f^{2(k)} + 2\lambda\omega_k^t)\right\}}\right], r = 1, \cdots, p.$$

According Theorem 9.16 in Amir (2017), the optimal iteration step size is

$$\alpha_T = \frac{\sqrt{2\Theta\left(\mathbf{w}^0\right)}}{L_f \sqrt{T+1}},$$

where $L_f \geq \|Z_\lambda'(\mathbf{w})\|_*$ for all $\mathbf{w} \in \boldsymbol{\Omega}$ for some $L_f > 0$, $\|\cdot\|_*$ is dual norm, and $Z_\lambda(\mathbf{w}) = -\mathrm{dMMD}_f^2{}^\top \mathbf{w} + \lambda \sum_{r=1}^p \omega_r^2$, $Z_\lambda'(\mathbf{w})$ is the first derivative of $Z_\lambda(\mathbf{w})$. Assume that and $\Theta\left(\mathrm{w}^0\right)$ satisfy

$$\Theta\left(\mathbf{w}^0\right) \geq \max_{\mathbf{w} \in \boldsymbol{\Omega}} B\left(\mathbf{w}, \mathbf{w}^0\right).$$

We consider $\ell_1$ norm as measurement of the sample space, thus $\|\cdot\|_* = \|\cdot\|_\infty$, where $\|\cdot\|_\infty$ is $\ell_\infty$ norm, and we have

$$\begin{aligned}\|Z_\lambda'(\mathbf{w})\|_* = \|Z_\lambda'(\mathbf{w})\|_\infty &\leq \max_{\mathbf{w} \in \boldsymbol{\Omega}} \left\{\max_r \left|-\widehat{\mathrm{dMMD}}_f^{2(r)} + 2\lambda\omega_r\right|\right\} \\ &\leq \max_r \left(\left|-\widehat{\mathrm{dMMD}}_f^{2(r)}\right|, \left|-\widehat{\mathrm{dMMD}}_f^{2(r)} + 2p\lambda\right|\right).\end{aligned}$$

Thus, let $L_f = \max_r \left(\left|-\widehat{\mathrm{dMMD}}_f^{2(r)}\right|, \left|-\widehat{\mathrm{dMMD}}_f^{2(r)} + 2p\lambda\right|\right)$. Since

$$\max_{\mathbf{w} \in \boldsymbol{\Omega}} B\left(\mathbf{w}, \mathbf{w}^0\right) = \max_{\mathbf{w} \in \boldsymbol{\Omega}} \sum_{r=1}^p \omega_r \log\left(\omega_r\right) = 0,$$

we can let $\Theta\left(\mathbf{w}^0\right) = 1$. If we consider $\ell_2$ norm as measurement of the sample space, we can obtain $\|\cdot\|_* = \|\cdot\|_2$, and the proof process is the same as under the $\ell_1$ norm. Finally, the optimal iteration is obtained, and we can also immediately obtain the optimal convergence rate. $\qquad\square$

