# OpenReview forum: "Regularized Maximum Mean Discrepancy for Variable Selection"
_ICLR.cc/2025/Conference — ICLR 2025 Conference Withdrawn Submission_

### Official Review · Reviewer_WeRj · 2024-10-31

**Soundness:** 3
**Presentation:** 3
**Contribution:** 2
**Rating:** 3
**Confidence:** 2

**Summary:**

This paper utilized the weighted norm with maximum mean discrepancy to solve the variable selection problem for two-sample testing, which is a very important problem in statistics and machine learning.

**Strengths:**

The overall procedure is sound. As far as I have checked, the proof is generally correct with detailed assumptions stated. Numerical study convince the soundness of the algorithm design.

**Weaknesses:**

Although the overall procedure seems to be sound, I have serious concerns regarding its theoretical foundation.
1. The authors proposed to solve Eq. (3), which is the objective combining empirical MMD statistics and $\ell_2$ regularization. Recall for least squares problem, $\ell_1$ regularization (lasso) helps with variable selection while $\ell_2$ does not. I believe the similar result applies here. So why does the author prefer $\ell_2$ instead of $\ell_1$ regularization?
2. For the evaluation of $\ell(\lambda)$ in Eq.(4) or Eq. (5), it involves the sample estimate of certain objective. So which type of data is utilized? The training data $\mathfrak{X}^{Tr}$ or testing data $\mathfrak{X}^{Te}$, or we need to perform training-validation split on $\mathfrak{X}^{Tr}$ and then utilize train-train data?
3. In the Algorithm the authors proposed to solve the optimization problem (3), which is unfortunately a non-convx problem in weight $\textbf{w}$. Then the authors resort to solve approximation optimization problem (6). What is the optimality gap between these? It is difficult for me to convince the effectiveness of this approximation algorithm.
4. Similarly, only the convergence analysis for solving the approximation optimization problem (6) is provided, but I cannot see the testing statistical power analysis when resorting to this approximation algorithm
5. Only the real data instead of the implementation code is provided for this paper.

**Questions:**

N/A

---

### Official Review · Reviewer_UAPb · 2024-10-31

**Soundness:** 2
**Presentation:** 3
**Contribution:** 3
**Rating:** 5
**Confidence:** 4

**Summary:**

This paper introduces a variable selection method using maximum mean discrepancy (MMD) under a binary-classification setting. The approach assigns and optimizes weights for each variable within a regularized MMD framework, setting some weights to zero for unimportant variables. These optimized weights act as an importance measure for identifying variables contributing to distributional differences. Simulations and real-data analysis demonstrates the empirical merit of this paper.

**Strengths:**

- Overall, this paper is clearly written.

- The paper presents an interesting approach to nonlinear variable selection within the MMD framework and develops a faster variant for its implementation.

**Weaknesses:**

- The term “Regularized MMD” in the title is somewhat ambiguous and does not fully capture the main idea of this paper.
- In [1], the simplex constraint yields a sparse estimator under the least squares setting, but it remains unclear if the solution to (2) demonstrates similar properties.
- The properties of $w_\lambda^*$ are not sufficiently investigated.
- Table 1: The statistical properties of both the original and accelerated methods should be included.
- The selection of benchmarked methods is limited [2-3]; there are numerous variants of (nonlinear) Lasso that could be considered.
- Type-I error studies should be integrated into the main text.
- An ablation study would help clarify whether using the ridge penalty is beneficial.

### Reference

- [1] Meinshausen, Nicolai. "Sign-constrained least squares estimation for high-dimensional regression." (2013): 1607-1631.

- [2] Ravikumar, Pradeep, et al. "Sparse additive models." Journal of the Royal Statistical Society Series B: Statistical Methodology 71.5 (2009): 1009-1030.

- [3] Yamada, Makoto, et al. "High-dimensional feature selection by feature-wise kernelized lasso." Neural computation 26.1 (2014): 185-207.

**Questions:**

- Why is the ridge penalty used instead of an $L_1$ penalty? Is there any theoretical justification for this choice?

- Can this workflow be extended to more general settings, such as regression with HSIC?

---

### Official Review · Reviewer_h4zi · 2024-11-02

**Soundness:** 2
**Presentation:** 3
**Contribution:** 2
**Rating:** 6
**Confidence:** 3

**Summary:**

This paper considers the problem of model selection when testing whether two distributions are identical. Traditionally, people calculate the Maximum Mean Discrepancy (MMD) between two distributions, and then perform test using MMD, namely reject the hypothesis that two distributions are identical if MMD exceeds a critical value. The authors observe that, unimportant variables may even hurt the performance of this MMD test. Therefore they proposed the weighted MMD, namely assign a weight to each variable. The optimal weights are chosen so that the weighted MMD is maximized. Variable selection can be done using these weights, facilitating the downstream two-sample test or binary classification. The authors also provide an optimization method with theoretical guarantee. Numerical experiments show the efficiency of the proposed algorithm.

**Strengths:**

The presentation of this paper is in general good, clearly stating the background and their intuition. This paper also provides comprehensive results regarding their proposed methods, including both theory for optimization and numerical simulations. Also the idea of using weighted MMD for model selection is novel.

**Weaknesses:**

My main concern is that, although the authors claim that weighted MMD is good for selecting variables, they do not provide theoretical guarantees that the optimal weight is beneficial for two-sample test or binary classification. Therefore the proposed method seems to be a little bit ungrounded.

**Questions:**

I am curious that whether it is possible to provide a theoretical guarantee that the optimal weight is beneficial for two-sample test or binary classification?

---

### Official Review · Reviewer_dxCu · 2024-11-03

**Soundness:** 2
**Presentation:** 2
**Contribution:** 2
**Rating:** 3
**Confidence:** 4

**Summary:**

The proposed method finds the importance levels of variables by optimizing a weighted version of maximum mean discrepancy (MMD). The objective function to minimize is negative weighted MMD plus a regularization term that imposes all variable weights to be $1$ when its parameter $\lambda$ is set to be infinitely large. The regularization parameter $\lambda$ is chosen through optimizing the performance of the task - two-sample test or classification. An approximate version of this method is presented to reduce the computational cost, obtained through a first-order Taylor expansion of the objective function. Consistency and convergence results are given for the solution of the approximate method. Experiments on two synthetic data sets and one gene expression data set show the advantage of the proposed method over the unweighted MMD for two-sample test, and the variable selection method by Lasso for classification.

**Strengths:**

* Proposition of a new method, presented along with theoretical and empirical results.

* The writing is clear and the paper is not hard to follow.

**Weaknesses:**

* The theoretical results only concern the consistency and the convergence of the approximate algorithm. There is no generalization or approximation guarantee.

* This work can benefit from a more extensive empirical study. For two-sample test, only one baseline is compared, and the experiments were exclusively conducted on synthetic data. For classification, two baselines are tested, on synthetic data and one real-word data set. Moreover, there is no error-bar in the reported empirical results.

* Most crucially, while the proposed method is claimed to be a method of variable selection, there is little comparison to other variable selection methods, expect some empirical results o compare the false discovery rate and the classification performance with the Lasso method.

**Questions:**

* The optimisation problem formulated in (4) does not seem to encourage sparse solutions. I think the weight vector obtained from (4) on a limited training set is generally non-sparse. How does this translate to a variable selection? Is there a thresholding procedure on the weight vector. If so, such procedure should be detailed in the article.

* Could the authors provide some approximation guarantees for the accelerated algorithm, or/and some generalization guarantee for the original method or its accelerated version?

* Why is there no error-bar in the experimental results? And why not use all the samples in the gene expression data set GSE2034 to obtain a large test set which can reduce the variance of the empirical performance, therefore allowing for a more reliable comparison?

* In Lines 262-264, it says "the training set, is used to compute the optimal weight vector $\hat w_{\hat\lambda}$ by optimizing the tuning parameter $\hat\lambda$", meanwhile in Lines 280-282 of Algorithm 1, it seems that $\hat\lambda$ is tuned in a cross validation manner by minimizing the loss $\ell(\lambda)$ on a test set. Could the authors clarity whether or not the tuning of $\hat\lambda$ involves exclusively the training set?

---

### Official Review · Reviewer_5kL2 · 2024-11-04

**Soundness:** 3
**Presentation:** 3
**Contribution:** 3
**Rating:** 5
**Confidence:** 3

**Summary:**

This paper proposes a variable selection scheme based on the maximum mean discrepany (MMD). The authors propose two variable selection algorithms, which are based on a regularized MMD objective, to assign weights to important variables and discard the unimportant ones. An accelerated scheme to compute the estimated weights is also presented. Theoretical results on the consistency of the estimated weights and the convergence of the accelerated scheme are provided. Experiments demonstrate the competitiveness of the proposed method to comparing methods.

**Strengths:**

- The paper is clearly motivated.
- The method is non-parametric and model-free, allowing a broad range of applications.
- Experimental results show sizable improvement over the original MMD framework

**Weaknesses:**

- The literature review could benefit from a broader references. For instance, other parameter selection schemes, other variants of MMD objective etc.
- The presentation in sec 2.4 is somewhat unclear. Equation (4) and (5) have an intricate structure. Readers could benefit from a clearer presentation/explanation.
- In the experiments, only two standard comparing methods are considered. The experimental results could be strengthened by including more advanced baselines.
- In the variable selection algorithms, randomness is involved. E.g., random subsets are chosen to compute the MMD value. It is generally recommended to report the mean and standard error over multiple random trials.

**Questions:**

- It seems that the first equation in sec 2.4 is obtained from (3) and the fact that the sum of omega is always a constant. In that case, (3) is equivalent to -MMD + \sum (omega - any constant)^2, not just one?
- There are multiple hyperparameters in the variable selection algorithms. How sensitive is the performance to the hyperparameters?
- Is the accelerated algorithm for computing the estimated weights similar to applying gradient descent to equation (3)?

---

### Note · Authors · 2024-11-28

I have read and agree with the venue's withdrawal policy on behalf of myself and my co-authors.